# Context-dependent change in the fitness effect of (in)organic phosphate antiporter *glpT* during *Salmonella* Typhimurium infection

Noemi Santamaria de Souza [1] ✉, Yassine Cherrak[1], Thea Bill Andersen [1], Michel Vetsch[1], Manja Barthel[1], Sanne Kroon [1], Erik Bakkeren [2], Christopher Schubert [1], Philipp Christen [1], Patrick Kiefer[1], Julia A. Vorholt [1], Bidong D. Nguyen[1] & Wolf-Dietrich Hardt [1] ✉

*Salmonella enterica* is a frequent cause of foodborne diseases, which is attributed to its adaptability. Even within a single host, expressing a gene can be beneficial in certain infection stages but neutral or even detrimental in others as previously shown for flagellins. Mutants deficient for the conserved glycerol-3-phosphate and phosphate antiporter *glpT* have been shown to be positively selected in nature, clinical, and laboratory settings. This suggests that different selective pressures select for the presence or absence of GlpT in a context dependent fashion, a phenomenon known as antagonistic pleiotropy. Using mutant libraries and reporters, we investigated the fitness of *glpT*-deficient mutants during murine orogastric infection. While *glpT*-deficient mutants thrive during initial growth in the gut lumen, where GlpT's capacity to import phosphate is disadvantageous, they are counter-selected by macrophages. The dichotomy showcases the need to study the spatial and temporal heterogeneity of enteric pathogens' fitness across distinct lifestyles and niches. Insights into the differential adaptation during infection may reveal opportunities for therapeutic interventions.

Enteric bacterial pathogens must continuously adapt to environmental challenges, including changes in nutrient abundance and challenges by the immune system, such as oxidative stress, to cause disease[1]. In the past decade, much progress has been made in discovering critical virulence factors that allow enteric bacterial pathogens to overcome specific obstacles (summarized in recent reviews[2–4]). Since the host's fitness depends on reducing damage by the pathogen, specific virulence factors face opposing selective pressures, where gene expression is advantageous in some contexts but detrimental in others. One example is the O-antigen, made up of glycans linked to the outer membrane of bacterial pathogens, which protects against membrane stress but is also recognized by intestinal antibodies[5–8]. Knowledge of this tradeoff has been leveraged to design vaccines that select for the loss of the O-antigen, which attenuated the pathogen[9]. Similarly, the expression of type-3-secretion systems (T3SS) is crucial to causing inflammation and invading host cells but reduces the pathogen's replication rate and increases susceptibility to membrane stress, which results in avirulent cells being selected[10–13]. Another example is motility conferred by flagellins, needed for enteric pathogens to reach, adhere to, and invade the epithelial cells that form the intestinal barrier[14,15]. Flagellin expression represents an Achilles heel for many enteric pathogens as the host's Toll-like receptor 5 selectively recognizes them

[1]Department of Biology, Institute of Microbiology, ETH Zürich, Zürich, Switzerland. [2]Sir William Dunn School of Pathology, University of Oxford, Oxford, UK. ✉e-mail: sanoemi@biol.ethz.ch; hardt@micro.biol.ethz.ch

and initiates an immune response[16]. As a result, the advantage of motility is thought to outweigh the risk of detection during early growth in the lumen. Once the epithelial barrier has been breached, facultative intracellular pathogens like *Salmonella enterica* serovar Typhimurium (*S.* Tm) can reside and grow inside host cells, where the disadvantage of flagella is thought to govern the selective pressure[17,18]. Understanding how a gene can increase or decrease a pathogen's fitness in a context-dependent fashion is important for developing effective strategies to control infectious diseases and understanding the evolution of pathogens with and without interventions such as antibiotic use or vaccination. For example, understanding that antagonistic pleiotropy affects the evolution of *S.* Tm's genes encoding surface-exposed structures helped explain why conventional vaccination attempts have been mostly unsuccessful (recently reviewed in refs. 19–22) On a positive note, the understanding allowed us to create a new form of vaccination termed evolutionary trap vaccine or EvoVax, which works by exploiting that escape variants are selected for by the immunized host immune system are attenuated[9,23].

So far, most examples of antagonistic pleiotropy have been attributed to the risk of detection by the host's immune system. Given that nutrient, mineral, and electron donor or acceptor availability differ considerably between the luminal and intracellular lifestyles, we hypothesized that some metabolic genes should similarly have fitness effects that are highly lifestyle-dependent and face opposing selective pressures. Evidence for antagonistic pleiotropy in metabolic genes adds to our understanding of the evolution of bacterial pathogens, with likely implications for preventative and therapeutic interventions.

The fitness conferred by a particular gene is determined by the context of the selective pressures, meaning that inactivating mutants are selected for in some niches, while the functional gene is selected for in others. The concept of such types of genes conferring opposing fitness effects is called antagonistic pleiotropy[24]. Antagonistic pleiotropy has been proposed as a driver of niche specialization[25,26]. A signature of antagonistic pleiotropy is the enrichment of loss-of-function mutations such as premature stop codons, compared to genes that are important for fitness in all niches[27]. A recent screen of over 100,000 *S.* Tm genomes found that the flagellin methylase gene *fliB*, which contributes to adhesion and host cell invasion, has four times as many premature stop codons as would be expected under neutral selection[27,28]. These mutations have likely been selected in natural environments rather than inside laboratories[27,28]. The increased frequency of premature stop codons suggests that strains proficient for the genes are favored in certain conditions, whereas certain niches select for inactivating mutations[27]. In addition to *fliB*, the screen identified a metabolic gene with an eightfold increased frequency of such premature stop codons: *glpT*[27]. The underlying selective pressures have remained unclear. One complementary approach to studying the fitness effects of particular mutations is to study the within-host selection of spontaneous pathogen mutants[29]. Recently, our lab studied within-host evolution using an *S.* Tm long-term murine infection model[13]. Strikingly, *glpT* mutations were found in 70% of isolates, including single nucleotide polymorphisms (SNPs) and deletions ("dels") at distinct locations (illustrated in Fig. 1a)[13]. This included cases where wildtype *S.* Tm prevented the re-establishment of a complex gut microbiota and a case where the microbiota was re-established by co-housing[13]. Moreover, the gene is not encoded in a genomic region that displays an unusually high frequency of mutations. Together, this indicates that these mutants were positively selected within the host (Fig. S1a). Our findings made in mouse models and the evidence in wild isolates support the existence of niches where

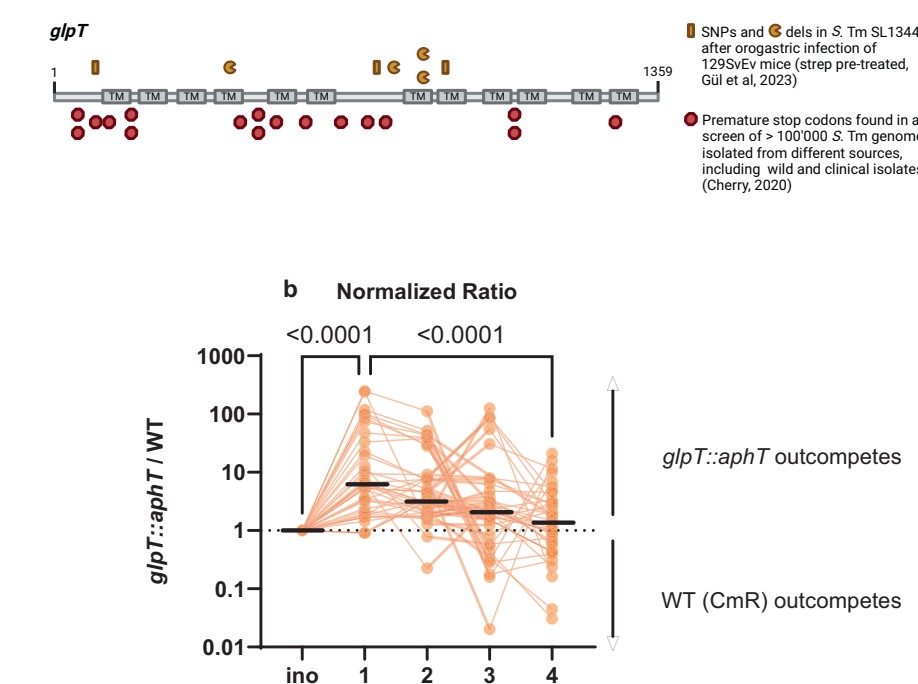

**Fig. 1 | Opposing selective pressures act on *glpT*. a** Single nucleotide polymorphisms and deletions found in the *glpT* gene of SL1344 after murine infection described in Gül et al.[70] PLoS Biology 2023 (upper half) and enriched premature stop codons found in sequences of different *S.* Tm isolates identified in Cherry, GBE, 2020 (lower half). Created in BioRender. Santamaria de Souza (2025) https://BioRender.com/f23o207 **b** The *glpT* mutant has a fitness advantage on the 1st day that gradually decreases over time. The data points show the aggregated competitive infection data of *glpT* vs. wild-type to highlight the data spread and the consistency of the change in the normalized ratio over time. The inoculum ("ino") has a normalized ratio of 1 since the mutant vs wild-type ratio is normalized to their ratio in the inoculum. The orange lines connect the data obtained from individual mice, whereas the black lines represent the median. The statistical significance was tested using a Kruskal–Wallis test. *n* = 40 mice. Source data are provided as a Source Data file.

the inability to express *glpT* confers a selective advantage, while other niches select for the wild-type (WT) *glpT* gene and that many *S. enterica* strains of different origins face opposing selective pressures acting for/against functional *glpT*. However, the nature of these selective niches has remained unclear.

*glpT* encodes for an antiporter of Sn-Glycerol-3-phosphate (G3P) and inorganic phosphate ($P_i$), across a concentration-gradient[27,30–32]. So far, there is very little information about the role of phosphate availability during enteric infection. Homologs of GlpT are found across all phyla, and highly conserved among *Enterobacteriaceae*, including several pathogens of global importance of the genera *Salmonella*, *Escherichia, Citrobacter, Shigella*, and *Klebsiella* (Fig. S1b)[33]. While GlpT has mostly been described for the import of G3P and export of $P_i$, we show that it can also import $P_i$. G3P is an essential building block of phospholipids and is thus found in every membrane. In addition, G3P can serve as an important carbon and or phosphate source for *S*. Tm and other mammalian pathogens, making the selection of *glpT* mutations even more surprising[31,34,35].

Here, we systematically assess the fitness defects of *glpT* mutants during distinct stages of orogastric mouse infection. We show that *glpT*-deficient mutants outcompete WT *S*. Tm during luminal growth when phosphate availability is high. In contrast, we demonstrate that the intracellular lifestyle in macrophages selects for a functional *glpT* gene. The lifestyle-dependent selection highlights the contextual fitness effect of *glpT*. Our findings emphasize the importance of antagonistic pleiotropy resulting from context-dependent metabolic trade-offs for pathogen evolution.

## Results

### The fitness of the *glpT* mutant changes during infection

To analyze the role of *glpT* in *S*. Tm gut-luminal growth, we compared the fitness of the *glpT* mutant to the isogenic WT strain. To this end, we infected streptomycin-pretreated mice with a 1:1 mix of a *glpT* knockout (KO) strain and the WT *S*. Tm SL1344 for 4 days (see Table 1; $10^7$–$10^8$ Colony Forming Units or CFUs, by orogastric injection). We determined the abundance of both strains in the feces of the mice by plating them on selective media and calculated the ratio between the strains, normalized to the ratio in the inoculum. By day 1 of infection, the *glpT* mutant outcompeted the WT, yielding a normalized ratio of -10 (Fig. 1b). Notably, the WT outcompeted from day 2 and the mutant/WT ratio progressively declined from around 10 to around 1 by day 4 p.i. (Fig. 1b). The data implies that the *glpT* KO strain has an advantage during the initial stages of gut colonization, whereas another selective pressure favors WT *S*. Tm during days 2–4. This provided us with the opportunity to investigate the cause of the change in the fitness effect of *glpT*.

### The fitness effect of *glpT* depends on substrate availability

GlpT is a G3P and $P_i$ antiporter with transport activities that depend on G3P and $P_i$ concentrations[36]. Because of that, we wondered if the competition between the *glpT* mutant and the WT depends on substrate availability[36,37]. To test that, we wanted to modulate the substrate availability to *S*. Tm in the intestinal lumen in three independent ways. First, we procured mouse chow with a reduced $P_i$ content of 0.1% as one way to reduce the phosphate availability to *S*. Tm in the intestinal lumen. We then compared the normalized ratio between the *glpT* mutant and the WT in mice fed the regular 0.8% $P_i$ chow to the normalized ratio between the *glpT* mutant and the WT in mice fed the 0.1% $P_i$ chow[36,37]. The diet with a reduced $P_i$ content significantly reduced the advantage of the *glpT* KO strain on day 1 p.i., to a normalized ratio of around 3 (Fig. 2a). This suggested that the fitness effect of *glpT* depends on the transport intensity and/or direction, which depend on substrate availability. As a second way to influence substrate availability, we supplemented 1% G3P in the drinking water of the mice (Fig. 2b). Similar to the reduction of $P_i$ availability (black data points,

Fig. 2a), this should lead to more G3P import and $P_i$ export, which should increase the fitness of the *glpT*-proficient WT. As expected, the G3P supplementation too significantly lowered the normalized ratio between the *glpT* knockout strain and the isogenic WT (Fig. 2b). As a third way to validate that the fitness effect of *glpT* depends on substrate availability, we knocked out all three known $P_i$ transporters encoded by *S*. Tm ($\Delta 3P_i$; *pitA, pstS, yjbB*)[38–41]. The resulting strain ($\Delta 3P_i$; *pitA, pstS, yjbB*) likely has less access to phosphate than the WT. While YjbB has been shown to be able to export $P_i$ when overexpressed, we did not know its transport activity at normal expression levels and in the absence of *pitA* and *pstS*[42,43]. Thus, we decided to take the conservative approach of removing it too, which should, if anything, reduce the effect of potential GlpT $P_i$ transport, if YjbB acts as a $P_i$ exporter in the tested conditions[41]. Next, we tested the relative fitness of the *glpT* KO in the $\Delta 3P_i$ background compared to the parental strain ($\Delta 3P_i$). Similarly to the low $P_i$ diet, the low availability resulting from this genetic background ($\Delta 3P_i$), strongly decreased the ratio between the *glpT* KO and its isogenic parent (WT background) to around 1 ($\Delta 3P_i$ background) (Fig. 2b). The decrease in the normalized ratio between the *glpT* KO and its parental strain, when access to $P_i$ was reduced with the modified diet, or the lack of $P_i$ transporters, establishes a link between the fitness effect of *glpT* and phosphate availability. The normalized ratio between the $\Delta glpT$ *pitA::cat*, $\Delta glpT$ *pstS::aphT*, and $\Delta glpT$ *yjbB::aphT* double knock-out strains can be found in Fig. S2a. We additionally tested the competitive fitness of the individual KOs (*pitA, pstS*, and *yjbB*, visualized in Fig. 2c) and the triple KO ($\Delta 3P_i$) in competition against the WT on day 1 p.i. All of them led to a nonsignificant ratio, close to 1, even when all three transporters were knocked out in the same strain (Fig. 2d). The neutral competition suggested that the luminal phosphate availability for *S*. Tm is surprisingly high.

We, therefore, decided to measure the total $P_i$ concentration in the cecum lumen of our mice. The median concentration was around 20 mM in the cecal content of untreated C57BL/6 mice and germ-free mice (Fig. S2b). We also quantified fumarate (detectable in lysed cells but not cecal content) and acetate (detectable in both) by targeted liquid chromatography/mass spectrometry to ensure that there was no contribution by the lysis of microbiota cells[44,45] (Fig. S2c). The range of phosphate concentration is in line with a recent study (40–50 mM $P_i$) that obtained mouse chow from the same provider[46]. The intestinal concentration of $P_i$ in humans is reported to reach 12 mM[47,48]. Such $P_i$ concentrations should result in a high $P_i$ availability, given that in vitro studies on the phosphate starvation response in *E. coli* and *S*. Tm use defined media with 0.5–1 mM $P_i$ as the non-limiting control[37,49,50].

### GlpT can import $P_i$ in vitro

While traditionally known for importing G3P and exporting $P_i$, the GlpT transporter is bidirectional and can additionally import Pi by the non-equimolar exchange of $P_i$ in *E. coli*[36,51]. Thus, $P_i$ import should be the main transport direction when the extracellular $P_i$ concentrations are considerably higher than G3P (visualized in Fig. 3a, scenarios 2 and 3)[31,36]. Because GlpT could directly contribute to $P_i$ import, we decided to test if the presence of *glpT* impacts intracellular $P_i$ availability in vitro. We constructed a reporter for $P_{pstS}$ that is highly sensitive to intracellular $P_i$ limitation[37,43,52]. All primers and plasmids used in this study can be found in Tables 2 and 3. We quantified the $P_{pstS}$-*nanoluc* reporter activity in strains with differing capacities of importing $P_i$: the WT, a *glpT* KO, a *pitA yjbB* double KO, and a *pitA yjbB glpT* triple knock-out strain. We did not knock out *pstS*, as it influences $P_{pstS}$ activity, and chose the nanoluc reporter gene since it is highly compatible for use in vivo, allowing us to use the same reporter in subsequent animal experiments (see Fig. S3b)[37,41,43,53]. As expected, the $P_{pstS}$ promoter was active in all strains when no $P_i$ was added to the medium (Fig. 3b). With the addition of $P_i$ to the media, the $P_{pstS}$ activity decreased with distinct cutoffs for the different strains. The additional *glpT* mutation in the

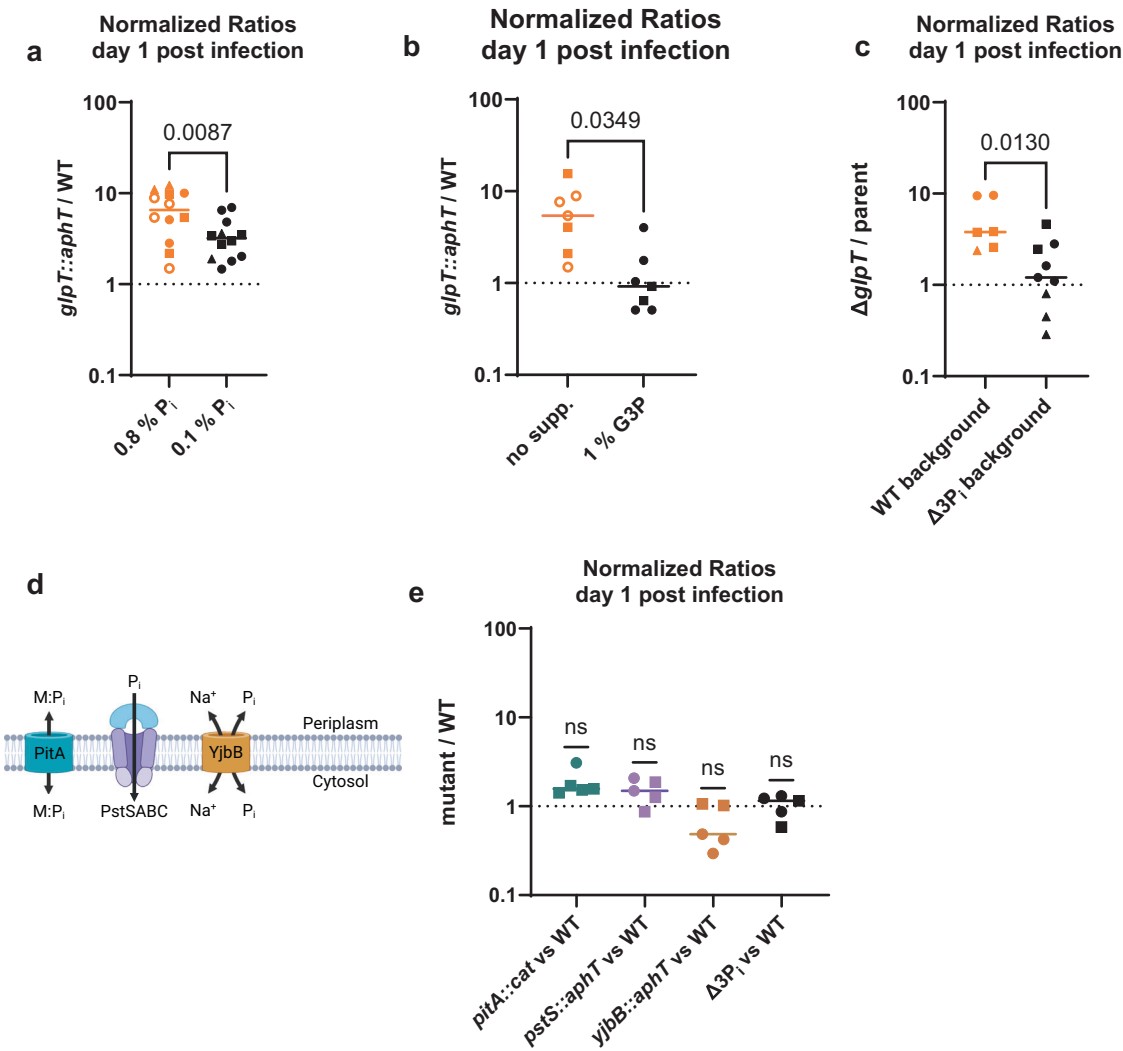

**Fig. 2 | The fitness effect of *glpT* is linked to the high phosphate availability in the murine cecal lumen. a** The fitness advantage conferred by the *glpT* mutant is reduced when mice are fed chow with an eightfold lower phosphate concentration (0.1%). Statistical significance was tested with a two-sided unpaired *t*-test. Lines represent the median. *n* = 12 mice, combined from four independent experiments, indicated by different shapes. Please note that four data points for the regular 0.8% $P_i$ chow control are also shown in panel 2b, as the same mice served as the control for both experiments for the sake of 3 R; these data points are shown as open circles. **b** The fitness advantage conferred by the *glpT* mutant is reduced when mice are fed 1% G3P via their drinking water. The statistical significance was tested with a two-sided unpaired *t*-test. Lines represent the median. *n* = 7 mice. The different shapes indicate they originate from two independent experiments. **c** The fitness advantage conferred by the *glpT* mutant is strongly reduced in the background of a strain deficient for the phosphate transporters *pitA*, *pstS*, and *yjbB* ($\Delta 3P_i$). The $\Delta 3P_i$

strain likely has less access to $P_i$, which should lower the intracellular $P_i$ concentration. The statistical significance was tested with a two-sided unpaired *t*-test. Lines represent the median. *n* = 7 mice for the wild-type background and *n* = 8 mice for the $\Delta 3P_i$ background combined from three independent experiments.
**d** Visualization of the three $P_i$ transporters encoded by *S*. Tm. Adapted from Bruna et al., Adv. in Exp. Med and Bio., 2022 for illustrative purposes. Created in BioRender. Santamaria de Souza (2025) https://BioRender.com/x62f158 (**e**) Competitive indices of phosphate transport mutants show that the $P_i$ concentration is high enough for mutants not to have a competitive disadvantage on day 1 post-infection. Significance was tested by comparing the median to the theoretical median of 1 using a one-sample Wilcoxon test (two-sided). Lines mark the median. *n* = 5 mice combined from two independent experiments. Source data are provided as a Source Data file.

*pitA*, *yjbB* deficient background changed the cutoff $P_i$ concentration fourfold (40 μM vs 10 μM for *pitA*, *yjbB*, Fig. 3b). These results suggest that GlpT can act as a $P_i$ importer in our *S*. Tm strain. GlpT's capacity for $P_i$-import was also reflected in the $OD_{600nm}$ at 24 h after dilution, which was lower for the triple KO ($\Delta pitA$ *yjbB*::*aphT* $\Delta glpT$) than the double KO ($\Delta pitA$ *yjbB*::*aphT*) strain when no $P_i$ was added to the medium (Fig. 3c). Thus, GlpT's transport activity depends on environmental parameters, such as the G3P and $P_i$ availability and GlpT might contribute to $P_i$ import during luminal growth, a milieu where we found phosphate availability to be high (Fig. 2d and S2a, b). The reduction in the fitness advantage for the *glpT* KO strain when $P_i$ availability is reduced (Fig. 2a, b) indicates that GlpT's additional $P_i$-import capability is harmful during day 1 p.i. luminal growth.

## G3P metabolism does not explain the d1 fitness advantage
To investigate if the luminal fitness advantage of the *glpT* knockout strain on day 1 p.i. is related to G3P metabolism, we constructed a library of isogenic knockout strains lacking genes encoding enzymes that have been implicated in glycerol metabolism (visualized in Fig. 4a)[31,54]. The WT and each isogenic mutant encode distinct fitness-neutral genetic barcodes, which allowed us to quantify the abundance of each strain in the pool by qPCR (visualized in Fig. 4b)[55,56]. UgpB is the substrate binding component of the UgpB-AEC₂ ABC transporter for G3P[57]. Since the *ugp* operon is induced during phosphate limitation[37,43], we expected to find a neutral normalized ratio between the *ugpB* KO and the WT strain (≈1, Fig. 4c), further indicating a high $P_i$ availability in the gut lumen. GlpF and GlpK enable the

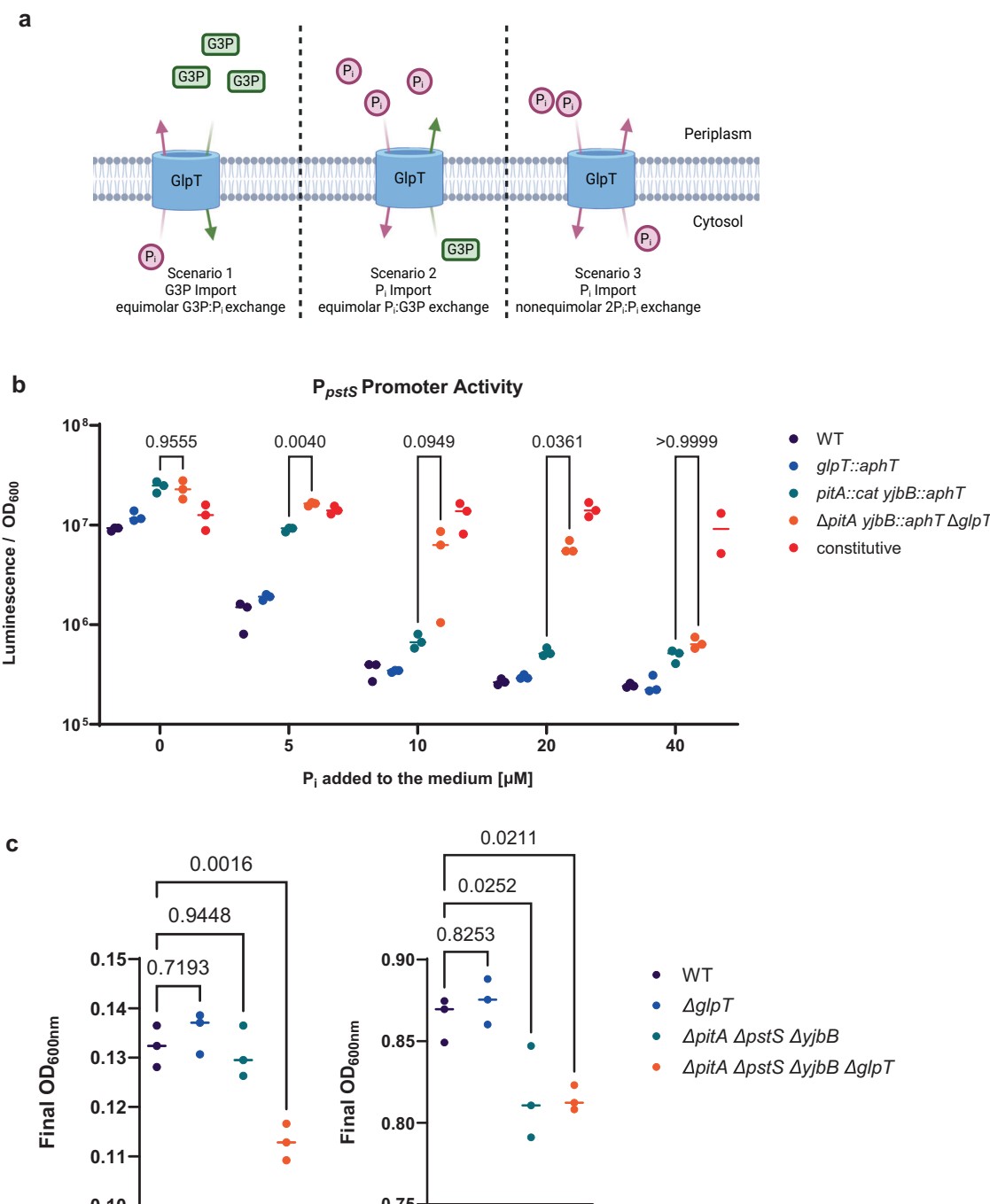

**Fig. 3 | GlpT can act as a phosphate importer. a** The three known transport modes of GlpT, depending on the availability of Glycerol-3-phosphate (G3P) and $P_i$. GlpT's G3P and $P_i$ antiport is bi-directional, in a concentration-dependent manner. In the first hypothetical scenario, there is more G3P than $P_i$ in the periplasm and the main transport direction in G3P import and $P_i$ export, by equimolar G3P:$P_i$ antiport. In the second hypothetical scenario, there is more Pi than G3P in the periplasm, and the main transport direction is $P_i$ import and G3P export, by equimolar $P_i$:G3P antiport. GlpT can additionally import $P_i$ by non-equimolar exchange of 2$P_i$:1$P_i$ in *E. coli*, which is shown in the third scenario. The different numbers of representative $P_i$ and G3P molecules aim at representing different periplasmic/cytoplasmic concentrations. Created in BioRender. Santamaria de Souza (2025) https://BioRender.com/

d47n170 (**b**) $P_{pstS}$ promoter activity as a readout for intracellular phosphate levels reveals that GlpT can increase intracellular phosphate levels. Cells were harvested after 24 h of incubation in a modified M9 medium. The statistical significance was calculated with an ordinary Two-Way ANOVA with Šidák multiple tests correction to account for the effect of the genotype and concentration. The lines represent the median. *n* = 3 biological replicates. **c** The final $OD_{600nm}$ after 24 h of incubation in a modified M9 medium shows that GlpT contributes to growth when phosphate availability and or uptake is limited. The lines represent the median. Statistical significance was tested using an ordinary one-way ANOVA with Dunnet correction. *n* = 3 biological replicates. Source data are provided as a Source Data file.

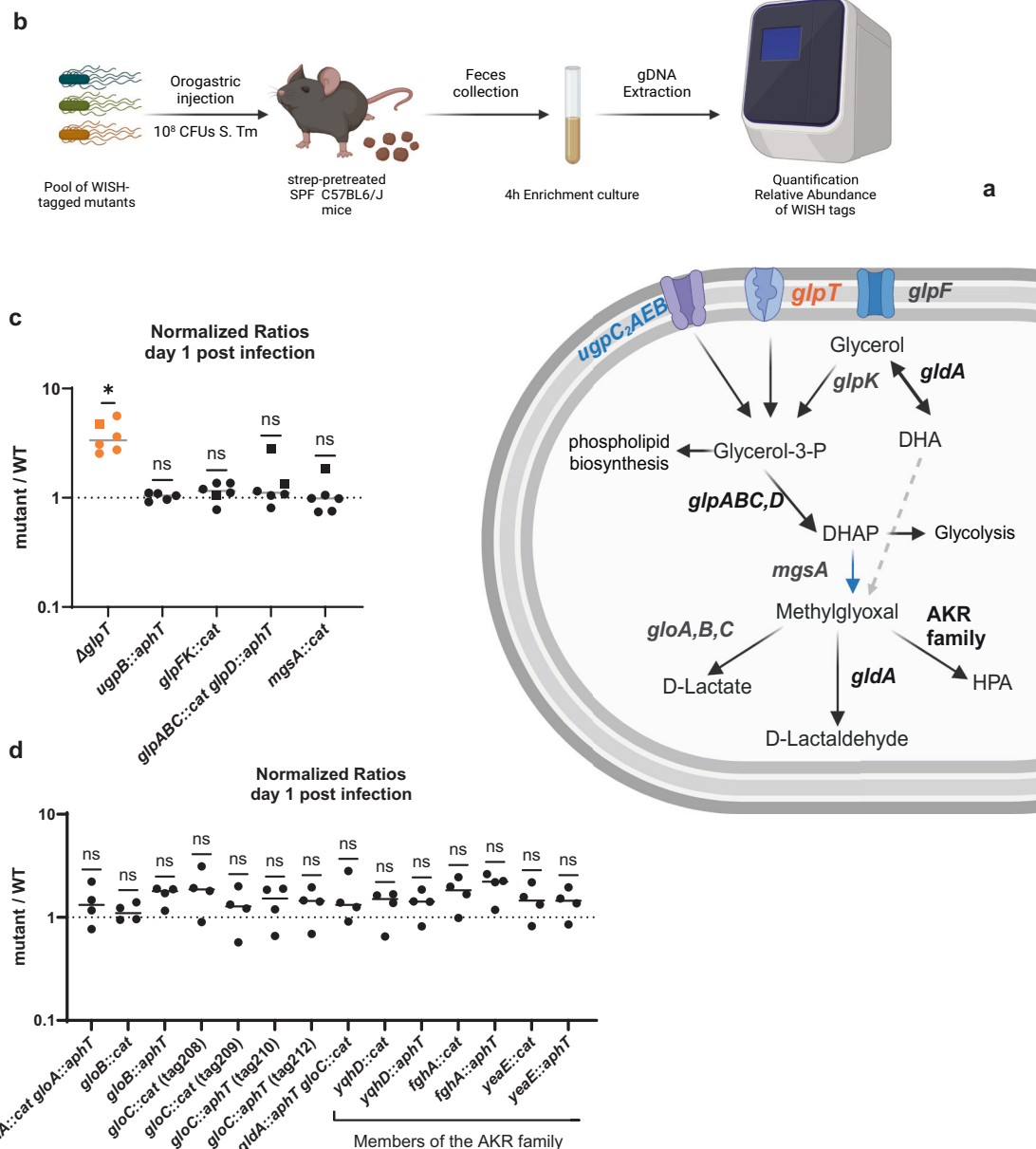

**Fig. 4 | There is no evidence for the glycerol pathway driving the competitive advantage on day 1 post-infection. a** Visualization of the glycerol metabolism in *S.* Tm. Mutants with a significant normalized ratio are highlighted in orange and genes or reactions affected by phosphate are highlighted in blue. Abbreviations: DHA(P) for Dihydroxyacetone (phosphate), AKR for Aldo-keto reductase, and HPA for Hydroxypropionic acid. The visualization was adapted from Tran et al.[122] and Subedi et al.[60] for illustrative purposes. Created in BioRender. Santamaria de Souza (2025) https://BioRender.com/v23o781 (**b**) Visualization of the setup of the screen to quantify the effect of the different players in the glycerol metabolism and methylglyoxal pathway on day 1 post-infection of streptomycin-pretreated C57BL/6 mice. Created in BioRender. Santamaria de Souza (2025) https://BioRender.com/ o63l097 (**c**) The normalized ratios for the mutants in the upper part of the glycerol metabolism pathway show that only the *glpT* mutation leads to a significant competitive advantage. The statistical significance was tested using the Wilcoxon signed rank test (two-sided) against a hypothetical mean of 1 for a neutral competition. Lines indicate the median. *n* = 6 mice, except for *ugpB::aphT n* = 5 mice. **d** The normalized ratios for the mutants defective in different parts of the methylglyoxal detoxification pathway show that the absence of detoxification enzymes does not lead to a fitness disadvantage. The statistical significance was tested using the Wilcoxon signed rank test against a hypothetical mean of 1 for a neutral competition. Lines represent the median. *n* = 4 mice. Source data are provided as a Source Data file.

glycerol uptake and phosphorylation, respectively[58]. The neutral ratio for the *glpFK* knock-out strain suggested that uptake via the GlpFK route does not influence the fitness in the gut lumen, potentially because GlpT could export the G3P[58]. The neutral ratio (≈1) for the two G3P dehydrogenases *glpABC* and *glpD* also suggested that there is little G3P conversion (Fig. 4c). We also analyzed *mgsA*, encoding for the methylglyoxal synthase, which catalyzes the formation of methylglyoxal, a strong electrophile with toxic effects[59,60]. The production of methylglyoxal is thought to be important during

phosphate limitation as it leads to the release of the phosphate group from DHAP, during high glycolytic flux, and oxidative stress[61,62]. Since MgsA is allosterically inhibited by $P_i$ binding (at 0.3–0.5 mM $P_i$ in *E. coli*), the high phosphate availability in the intestinal lumen probably inhibits MgsA (Fig. 4c)[59,60,63]. The lack of methylglyoxal production was supported by the competition with the mutants deficient in enzymes involved in methylglyoxal detoxification (*gldA, gloA, gloB, gloC, yqhD, fghA, yeaE*), which were not outcompeted by the WT on day 1 p.i. (Fig. 4d)[60,64,65].

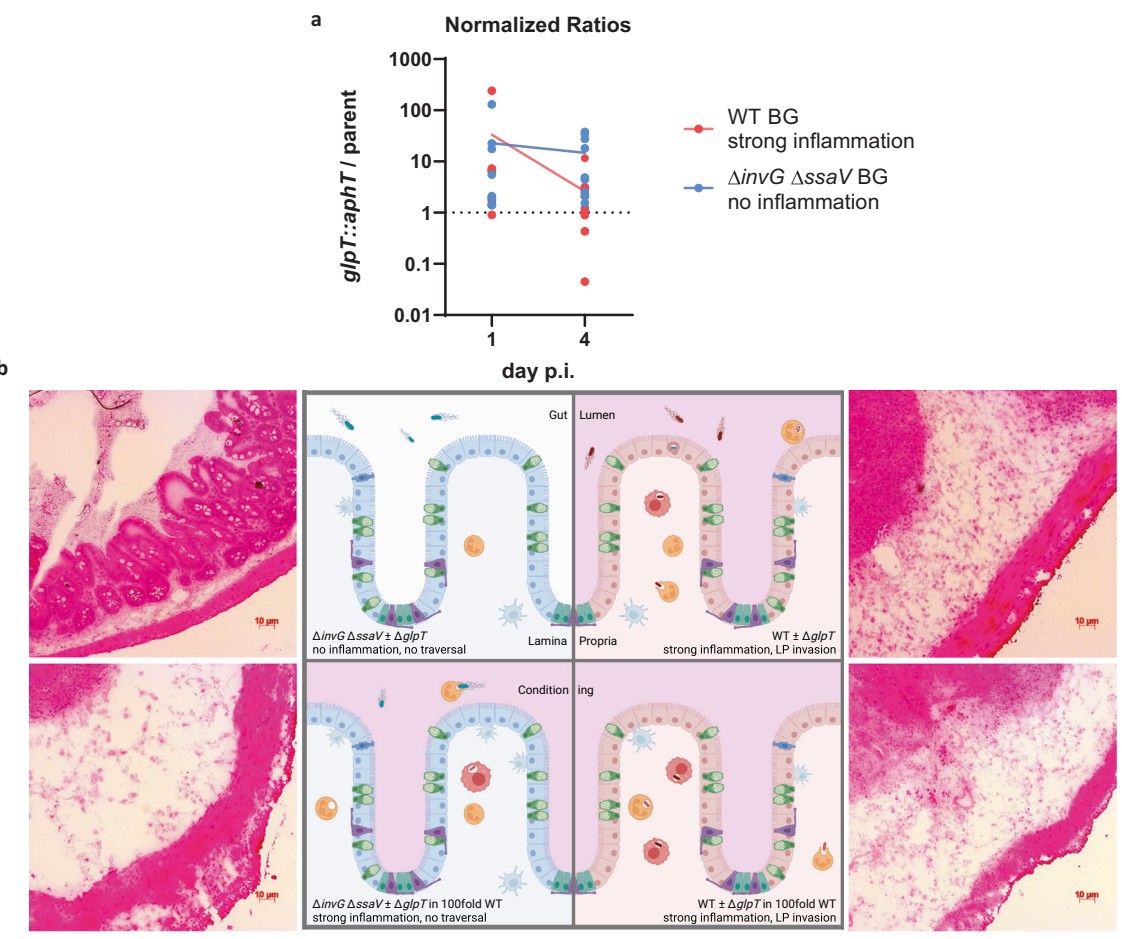

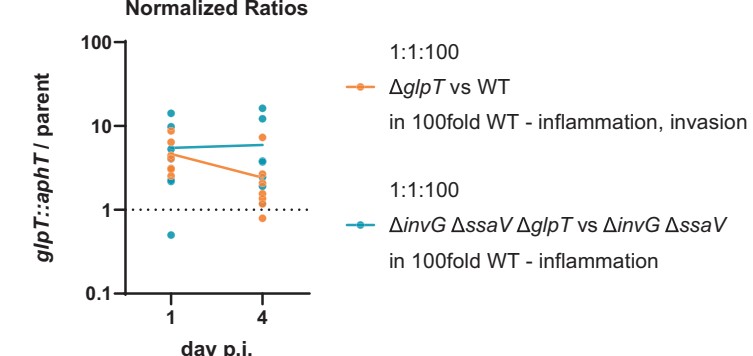

**Fig. 5 | A niche accessible to T3SS-proficient *S*. Tm selects for *glpT*. a** The normalized ratio only decreases when the strains are proficient for the two type three secretion systems and, therefore, virulent. The lines connect the means. *n* = 9 mice in three individual experiments. **b** Visualization of the setup and effect on *S*. Tm's capacity to cause inflammation and or invade the lamina propria. Representative microscopy images (10×) of cecum cryosection demonstrate the absence of inflammation on the top left and strong inflammation in all other setups.

Illustration created in BioRender. Santamaria de Souza (2025) https://BioRender. com/x00c911 (**c**) Triple infection experiment, with a wild-type strain that leads to high inflammation levels in both groups. The normalized ratio only decreases when the strain is proficient for the type three secretion systems, enabling access to the lamina propria and intracellular replication. The lines connect the means. *n* = 7 mice in three individual experiments. Source data are provided as a Source Data file.

## The *glpT* mutant is outcompeted by day 4 of infection

Since the ratio between the *glpT* KO and the WT declines around 10-fold between days 2–4 p.i. (Fig. 1d), we considered that the luminal phosphate availability might decrease in the presence of inflammation. From around day 2 p.i. inflammation causes a dramatic shift in the chemical composition of the gut lumen, which substantially influences

the fitness of enteric bacteria. Therefore, we examined the ratio of a *glpT* mutation in an Δ*invG* Δ*ssaV* background, a strain deficient for *S*. Tm's T3SS and incapable of eliciting pronounced gut inflammation[66,67]. In this strain background, the *glpT* mutant also had a fitness advantage by day 1 p.i. (ratio ≈ 10; Fig. 5a, shown in blue). Remarkably, the ratio of the *glpT* mutant did not decrease in the Δ*invG* Δ*ssaV* background on

subsequent days (Fig. 5a, shown in blue). However, when we quantified the $P_i$ concentration in mice infected with WT *S.* Tm, we saw no difference between day 1 and day 4 p.i. (Fig. S3a). We additionally measured the activity of our $P_{pstS}$-*nanoluc* reporter (as a readout for intracellular $P_i$) and found no difference between day 1 and 4 p.i. in WT or Δ*invG* Δ*ssaV* infected mice (Fig. S3b). Despite the supplementation of phosphate (100 mM $K_2HPO_4$) in the drinking water of the mice, we observed no difference in their luminal concentration or the ratio of the *glpT* mutant (Fig. S3c, d). This suggested that there was no significant decrease in luminal $P_i$ availability.

We additionally wanted to exclude that the transcription of *glpT* changed over time. To do so, we created a $P_{glpT}$-*gfp* transcriptional reporter and quantified the gfp+ fraction of cells stained with an anti-O5 LPS antibody to detect all *S.* Tm cells using a gating strategy shown in Fig. S4a. We ensured the plasmid was maintained over the infection without any significant fitness cost (Fig. S4b–d) and observed no significant difference in $P_{glpT}$ activity between day 1 and day 4 p.i. (Fig. S4c). These observations suggested that the murine gut luminal niche selects for *glpT* mutants and that other niches within the mouse may select against *glpT* mutants by days 2–4 p.i. (Fig. 1b). The nature of such niches remained unclear.

### A T3SS-accessible niche selects against the *glpT* mutant

In addition to the inflammation state of the cecum, the expression of functional T3SS enables *S.* Tm to take up its intracellular lifestyle and form gut tissue reservoirs[68,69]. Thus, the declining fitness of the *glpT* mutant detected from day 2 and onwards might result from the selection of the WT over the *glpT* knockout when the pathogen resides in intracellular compartments within host tissues. To discriminate between the effect of inflammation on the luminal subpopulation and a potential effect on the intracellular subpopulation, we set up an experiment in which the same two sets of strains competed against one another (*glpT::aphT* vs. WT and Δ*invG* Δ*ssaV glpT::aphT* vs. Δ*invG* Δ*ssaV*). However, this time, we diluted the mix of the two competing strains in a 100-fold excess of untagged WT *S.* Tm. The excess of WT *S.* Tm leads to pronounced gut inflammation in both setups (visualized in Fig. 5b)[70]. As anticipated, in the competition between *glpT* mutant and the isogenic WT strain proficient for gut tissue invasion, the *glpT* mutant outcompeted the WT on day 1 p.i., while the ratio declined to around 1 by day 4 p.i. (Fig. 5c, shown in orange). In contrast, in the competitive infections between Δ*invG* Δ*ssaV glpT::aphT* vs. Δ*invG* Δ*ssaV* strains in the inflamed gut that cannot invade the gut tissue, the *glpT* mutant outcompeted throughout the infection (ratio≈7, Fig. 5c, shown in turquoise). This indicated that the selection of the WT is not due to inflammation-associated changes in the cecal lumen milieu. Instead, it seems that a niche mainly accessible to cells with functioning T3SS is selected for the WT.

### The *glpT* mutant has a replication deficit inside macrophages

In the lamina propria, many *S.* Tm cells are lodged within macrophages and *S.* Tm can replicate in tissue-resident macrophages, a $P_i$-scarce environment[52,71]. We considered that WT *S.* Tm could benefit from *glpT* expression in this environment. To test that, we infected Raw264.7 macrophages either with the WT or the *glpT* mutant and quantified *S.* Tm's replication rates (by dividing the CFUs quantified for 24 h p.i. by the CFUs for 2 h p.i.). We found that the *glpT* mutant has a replication deficit (Δ*glpT* empty vector) that can be rescued by plasmid complementation (Δ*glpT* $P_{nat}$-*glpT*, Fig. 6a). To test if the activation state of macrophages matters, we also quantified the replication rate in Raw264.7 that were primed with IFN-γ prior to infection. The priming with IFN-γ slightly increased the importance of *glpT* for intra-macrophage replication (Fig. 6b). In both cases, the complementation plasmid was stably maintained without antibiotic selection (Fig. 6c). Since host cells can limit the amount of phosphate available to intracellular pathogens when they are infected, we hypothesized

that the effect of the *glpT* mutation on intra-macrophage replication should not depend on the presence of the three $P_i$ transporters *pitA*, *pstS*, and *yjbB* since $P_i$ availability is already limited in the niche[72–74]. To test that, we compared the replication rate of the *glpT* mutant in the WT background to the ratio in the Δ3$P_i$ (*pitA*, *pstS*, and *yjbB* deficient) background. We found that the *glpT* mutation significantly attenuated intracellular replication in both cases (Fig. 6d). We also hypothesized that with the lower $P_i$ availability, the entire glycerol pathway should be active inside macrophages. Since the methylglyoxal pathway was recently shown to contribute to the resistance against oxidative stress generated by macrophages, we decided to investigate if *glpT* contributes to the methylglyoxal pathway inside macrophages[62]. To this end, we constructed a transcriptional reporter for $P_{gloC}$, which regulates the expression of a type II glyoxalase involved in the last step of the methylglyoxal pathway (see Fig. 4a)[75]. The higher $P_{gloC}$ promoter activity for the WT inside Raw264.7 macrophages suggests that *glpT* contributes to the methylglyoxal pathway and can, consequently, contribute to the resistance against the oxidative stress challenge faced by *S.* Tm with an intra-macrophage lifestyle (Fig. 6e)[62]. To discriminate between a methylglyoxal-dependent effect, such as a contribution to resistance against oxidative stress, and a methylglyoxal-independent effect, such as import of G3P as a carbon source, we compared the replication rate of the *glpT* deficient strain, to the replication rate of strains deficient for only *mgsA*, *gloA*, and *gloB* or in combination with the *glpT* knock out. MgsA is responsible for the formation of methylglyoxal, while GloA and GloB are involved in its detoxification (see Fig. 4a)[59,60,64,65,76,77]. *mgsA*, *gloA*, and *gloB* deficient strains should all be less protected against oxidative stress than WT *S.* Tm[62]. The replication rate was similar for all strains (around 1, Fig. S5). This indicated that our replication assays do not have sufficient resolution to be able to disentangle GlpT's contribution to replication in Raw264.7 macrophages in methylglyoxal-dependent and/or -independent ways, even when we combined datasets from three separate experiments, to increase the statistical power (Figure S5)[64,76,77].

### Macrophages tip the balance between the *glpT* mutant and the WT

Finally, we tested if intramacrophage replication could impact the ratio between the *glpT* mutant and the WT in fecal samples. Since 99.999% of the luminal population is killed by neutrophils on day 2 p.i., *S.* Tm that have replicated inside macrophages could contribute to the recolonization of the intestinal lumen[78–80]. Therefore, the higher intramacrophage replication rate of the WT compared to the *glpT* mutant might shift the balance towards a neutral ratio (≈1) in fecal samples by the last days of infection. To test that, we injected mice with an anti-CSF1R blocking antibody that selectively reduces the precursors of macrophages in the lamina propria of the mice[81–83]. As a control, we intraperitoneally injected mice with PBS or an isogenic antibody control. As expected, the ratio between the *glpT* mutant and the WT was neutral for the mice injected with PBS or with the isogenic antibody control (Fig. 6f). Strikingly, in the mice treated with anti-CSF1R, the *glpT* mutant consistently outcompeted the WT (Fig. 6f). This provided additional evidence supporting that the macrophages select *S.* Tm cells expressing *glpT*.

### Discussion

Previous studies have shown that classical virulence factors face opposing selective pressures, which have been leveraged as a therapeutic strategy[9]. In this study, we investigated whether the distinct niches encountered by the enteric pathogen *S.* Tm lead to opposing fitness effects and selective pressures on metabolic genes. We characterized *glpT* as a candidate metabolic gene that faces opposing selective pressures. We found that it has a lifestyle-dependent fitness effect, where *glpT* is harmful during luminal growth, characterized by a high phosphate availability. However, the same gene is important for

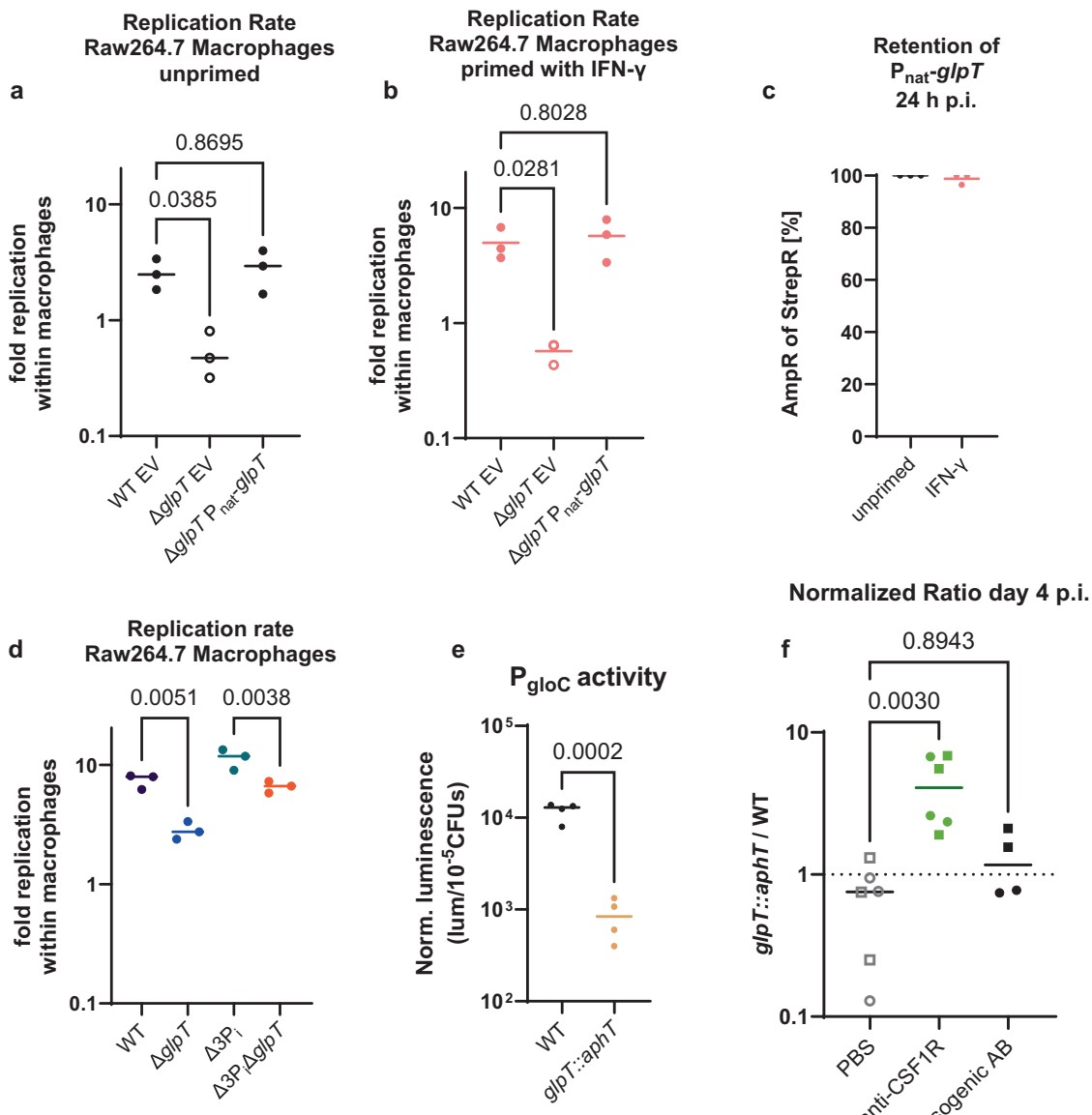

**Fig. 6 | GlpT is important for intracellular replication inside macrophages.**
**a** Replication rate inside infected Raw264.7 macrophage cells as CFUs at 24 h post-infection divided by the CFUs at 2 h post-infection after a gentamicin protection assay. The statistical significance was tested by an ordinary one-way ANOVA with Dunnet correction. The lines indicate the median. $n = 3$ Raw264.7 biological replicates. **b** Replication rate inside infected Raw264.7 macrophage cells. The Raw264.7 macrophages were primed with 2 ng/mL IFNγ for 24 h prior to infection. The lines indicate the median. The statistical significance was tested by an ordinary one-way ANOVA with Dunnet correction. $n = 3$ biological replicates (except for the empty vector condition where $n = 2$). **c** Plasmid retention for the experiments shown in panels a and b was quantified for the *glpT* mutant carrying the $P_{nat}$-*glpT* complementation plasmid by replica plating from LB Agar containing Streptomycin onto LB Agar with Ampicillin and then Streptomycin. The percentage of cells resistant to Ampicillin and Streptomycin is shown. The lines indicate the mean and the whiskers and the smallest and largest values, respectively. $n = 3$ agar plates.

**d** The replication rates in Raw264.7 macrophage cells show that *glpT* is important for replication regardless of whether the strain is proficient for the canonical $P_i$ transporters. The statistical significance was tested by an ordinary one-way ANOVA with Šidák correction. $n = 3$ biological replicates. **e** The $P_{gloC}$ promoter activity suggests that the methylglyoxal pathway is less active in the *glpT* mutant. The lines indicate the median. The statistical significance was tested with a two-sided unpaired *t*-test. $n = 4$ biological replicates. **f** Normalized ratio between the *glpT* mutant and the wild-type in mice intraperitoneally injected with PBS (open symbols), anti-CSF1R (green), or an isogenic antibody control (black). The different shapes (circles and squares) indicate the origin of two independent experiments and the lines indicate the median. In the absence of macrophages, the *glpT* mutant continues to outcompete the wild-type on day 4 post-infection. The statistical significance was tested using an ordinary one-way ANOVA with Dunnet correction. $n = 6$ or 4 (isogenic antibody) mice. Source data are provided as a Source Data file.

intracellular replication in Raw264.7 macrophages, and the wild type is selected in the presence of macrophages in vivo.

The availability of sugars, lipids, and metals has been shown to impact the fitness of *S.* Tm and other enteric pathogens during luminal growth[1,84]. However, it was unclear what the impact of phosphate availability on *S.* Tm's luminal growth would be. Since the $P_i$ concentration impacts GlpT's transport and glycerol utilization, we

considered phosphate availability during luminal growth. We found that there is enough $P_i$ in the cecal lumen (~20 mM, Fig. S2b) for *S.* Tm to stay fit (neutral ratio between KO and WT) even when we deleted its canonical $P_i$ transporters (*pitA*, *pstS*, and *yjbB*, Fig. 2b–d and S2b). PitA and PstS are believed to be more efficient at importing $P_i$ than GlpT, during in vitro growth[85–87]. Interestingly, the effect of *glpT* on luminal growth depends on phosphate availability, as the selection of the *glpT*

KO is diminished when mice are fed a low-phosphate diet or when the strain is deficient for its canonical $P_i$ transporters (Fig. 2a, b), suggesting that *glpT* is only harmful when $P_i$ availability is high. While nutrient limitation has been tied to protection against enteric pathogens, the negative effect of high phosphate availability demonstrates that an excess can also be harmful[88]. While GlpT has mostly been described for the import of G3P in exchange for $P_i$, the reverse transport direction was discussed several decades ago[36]. Additionally, GlpT has been shown to be able to import by non-equimolar $P_i$ exchange in *E. coli* (visualized in Fig. 3a)[36,51]. Our $P_{pstS}$ reporter data indicates that in vitro, GlpT can also import $P_i$ in *S*. Tm and, thus, directly contribute to the intracellular $P_i$ concentration (Fig. 3). The high luminal phosphate availability and capacity for GlpT to import $P_i$ is supported by the neutral fitness effect of knocking out the other genes involved in glycerol metabolism on day 1 p.i. (Fig. 4). GlpT's $P_i$ import may reduce WT *S*. Tm growth in the intestinal lumen and explain the fitness advantage of the *glpT* KO during luminal growth.

The luminal disadvantage for *glpT* expression could have multiple non-exclusive reasons. If $P_i$ import by GlpT is coupled to G3P export, the reduction in the important metabolite G3P could reduce fitness. In addition, the $P_i$ import in a niche with such a high phosphate availability could also pose a risk for phosphate intoxication. Although phosphate intoxication has not previously been investigated in the context of luminal colonization, there is precedent for *S*. Tm's phosphate intoxication during in vitro growth[41]. The mechanism for the $P_i$ intoxication proposed by the authors is that phosphate chelates magnesium ions, which prevents magnesium-dependent processes, including translation[89]. Magnesium chelation seems a severe risk for *S*. Tm as it evolved a process that inhibits its own $F_1F_O$ ATPase to free $Mg^{2+}$[90]. Moreover, increased intracellular $P_i$ concentrations directly reduce the Gibb's free energy for ATP hydrolysis under physiological concentrations and, thus, the concentration of ATP in the cell[91].

It is currently unclear if expressing *glpT* in the lumen is only disadvantageous or if it prepares the WT for other niches. Importing $P_i$ via GlpT could be beneficial for two reasons. Firstly, to prevent the accumulation of G3P, which, like other sugar phosphates, can be growth-inhibiting[92]. Secondly, it could be a means to accumulate $P_i$ since PitA does not import much $P_i$, and the Pst complex is repressed by intracellular $P_i$[37,43]. The 70% reduced mortality rate in mice infected with *S*. Tm mutants that cannot store $P_i$ in the form of polyphosphate suggests that phosphate storage greatly affects its virulence[93]. Since $P_i$ levels are limited at several sites of infection, such as the blood and intracellular niches, many pathogens might benefit from storing phosphate while they grow in a high phosphate environment, such as the intestinal lumen. Access to phosphate at $P_i$-limited infection sites seems to affect virulence as sepsis patients with higher serum phosphate levels have a higher mortality[94,95]. Given its potential for therapeutic intervention, it would be interesting to explore the links between phosphate storage during luminal growth and virulence in future experimental work[71].

In clinical contexts, *glpT* mutations are also selected by treatment with Fosfomycin and related antibiotics that have been proposed as a treatment for infections with multidrug-resistant Enterobacteriaceae[30,96,97]. Thus, it might be important to prescribe Fosfomycin in combination with another antibiotic to prevent the selection of *glpT* mutations that could lead to a dangerous short-time expansion of enteric pathogens in the gut lumen, especially in immunosuppressed patients who might not be able to mount an immune response that could limit *glpT* mutants.

In the second part, we examined the striking change in the selection pressure acting on the *glpT* KO on days 2–4 p.i. when the WT outcompetes the *glpT* KO. We found that this selection of the WT depends on *S*. Tm's two T3SS, which enable the invasion and replication inside host cells[98,99]. The experiments with Raw264.7 cells showed that *glpT* is important for replication inside macrophages. When we depleted macrophage populations in vivo, the *glpT* KO outcompeted

the luminal population on day 4 p.i. The correlation between the presence of *glpT* and $P_{gloC}$ activity suggested that the entire pathway is active inside the phosphate-limited macrophages. This suggests that *S*. Tm imports G3P in this niche and that the methylglyoxal pathway is not inhibited by $P_i$. The importance of GlpT inside macrophages could be linked to its contribution to the methylglyoxal pathway. Detoxification of methylglyoxal has recently been shown to be crucial for *S*. Tm's survival of the oxidative stress generated by macrophages[62]. Additionally, G3P metabolism could be an important carbon source for intracellular replication, given the niche's nutrient and mineral scarcity[100]. G3P metabolism entails less phosphorylation than the utilization of glucose, as it enters the glycolytic pathways at a midway point. Such a benefit for G3P metabolism was recently shown for another pathogen that can replicate inside phagocytic host cells, *Listeria monocytogenes*, in J774.7 macrophages[101]. Furthermore, G3P metabolism has been found important for *S*. Tm colonizing the spleen of mice, a site where the majority of *S*. Tm reside in red pulp macrophages[35,102].

An interesting side observation was the heightened replication rate of the Δ3Pi strain in Raw264.7 macrophages (Fig. 6d). The finding was in line with a prior publication, which found that a *phoB* deficient *S*. Tm strain showed a higher replication rate in J774 macrophages[103]. A similar conclusion can be drawn for replication in HeLa cells, where phosphate availability was inversely correlated with *S*. Tm's replication rate[74]. The authors knocked out the eukaryotic PIT1 gene, which should increase phosphate availability to *S*. Tm, and saw a reduced replication rate for *S*. Tm[74]. Accordingly, overexpression of PIT1 increased *S*. Tm's replication rate[74]. In contrast, a different study found that a *phoB*- and *pstSCAB*-deficient *S*. Tm strain was attenuated in replication in HeLa cells and Raw264.7 macrophage cells[52]. The conflicting results show that the effect of phosphate availability and lack thereof at intracellular sites on *S*. Tm fitness is unresolved and should be addressed in future studies. Since the luminal *S*. Tm population is decimated by neutrophils that kill 99.999% of luminal cells on day 2 p.i., we have suspected that re-seeding events from intracellular sites contribute to the re-plenishing of the luminal *S*. Tm population after the luminal population has been decimated by neutrophils or antibiotic treatment[78,104]. The contribution of intramacrophage replication to the recolonization of the lumen would be in line with a study that used an engineered (STm^CytoKill) *S*. Tm strain to show that cytosolic replication in enterocytes significantly contributed to luminal expansion and fecal shedding[79]. Here, we saw that the *S*. Tm cells expressing *glpT* outcompeted *glpT* deficient cells in the intestinal lumen when macrophages were present as an intracellular niche in the host tissue. The change in ratio in the intestinal lumen by 4 p.i. suggests that the higher replication rate of *glpT*-expressing cells inside macrophages promotes re-colonization of the intestinal lumen. Overall, this selection for WT *glpT* could explain why this gene is evolutionarily conserved.

The evolution dynamic of *glpT* is particularly interesting given that enteric pathogens infecting via the fecal-oral route reach the gut lumen before establishing an intracellular lifestyle. The number of cells and replication rate are higher in the lumen, where *glpT* mutants are selected. *glpT* mutations could, therefore, represent an example of antagonistic pleiotropy, where a mutation is selected during an (early) context but later results in a disadvantage. A famous example of antagonistic pleiotropy is *E. coli*'s resistance to phage T4 infection, which reduces its fitness in the absence of the phage[105,106]. Alternatively, the selection of *glpT* mutants observed in our study may result from reduced levels of colonization resistance, which are characteristic of the mouse models that we have used[107]. This should enhance the chances of selecting for mutants with enhanced fitness in such low-colonization-resistance niches. Transient alleviation of colonization resistance by food, antibiotics or other stressors may alter the gut-luminal niche to select for mutations in particular genes including in natural *Salmonella* populations, explaining why the loss of function

of *glpT* and several dozen other genes are selected often enough to be picked up in when analyzing 100,000 genomes[27]. Positive selection of *glpT*-deficient mutants is only possible when the *glpT* gene is expressed by the parental strains in the respective niches.

Context-dependent selection for mutants has also been observed for other enterobacteriaceal genes. The expression of the virulence regulator *hilD* of *S*. Tm is one prominent example[12,13]. Here, costs of expression in the gut lumen result in the selection of *hilD*-inactivating mutations when the pathogen grows over extended periods of time at high densities within the lumen of a microbiota-depleted inflamed gut. Similarly, gut luminal growth of *E. coli* K12 in germ-free mice selected for mutants in flagella, *ompB* and the maltose regulator gene *malT*[108]. Another study found that gut-luminal growth can select for the co-existence of *E. coli* strains, which can utilize galactitol, and *gat* mutants, which cannot[109]. Bioinformatics analysis of carbohydrate utilization genes in *Enterobacteriaceae* provided evidence that such genes might be under diverse context-dependent selection pressures, resulting in substantial differences in the control of carbohydrate utilization gene expression between different *Enterobacteriaceae*[110]. In conclusion, our data indicates that *glpT* mutations are selected in the gut luminal niche, resulting in a disadvantage in the phagocytic and potentially other phosphate- and nutrient-limited niches. The counterselection follows much slower kinetics, as the ratio reaches 10 within the 1st day of infection but takes the following 3 days to decline to the same extent. Yet, the data suggest that the capacity for intramacrophage replication has a crucial effect on the luminal population and, by extension, fecal shedding and infection of new hosts.

In conclusion, the fitness gains and losses of *glpT* show that opposing selective pressures can indeed act on metabolic genes. The example highlights how the heterogeneity between the luminal and intracellular niches results in lifestyle-dependent selection, even for a single gene within the same host. Similar dynamics likely exist in other facultative intracellular pathogens. The results reveal a need for studying the dynamic heterogeneity in pathogen populations at a higher resolution. In addition, such metabolic tradeoffs reveal an opportunity for therapeutic intervention, as past studies have shown that environmental changes during infection impose severe bottlenecks on pathogens. Consequently, the heterogeneity in selective pressures, need for adaptation and tradeoffs represents a unique opportunity to identify therapeutic targets, especially for the growing number of bacterial pathogens that cannot efficiently be treated with existing antibiotics.

# Methods

## Ethics statement
All animal experiments conducted in this study were reviewed and approved by Tierversuchskommission, Kantonales Veterinäramt Zürich, under the license numbers ZH158/2019, ZH108/2022, and ZH109/2022 and complied with the cantonal and Swiss legislation. We hold publication rights for all figures displayed in this study.

## Mouse infections
Male and female mice aged between 8 and 12 weeks were randomly assigned to experimental groups. All mice are bred and housed in individually ventilated cages in SPF and *E. coli*-free conditions at the EPIC facility at ETH Zürich. Germ-free mice were bred in flexible film isolators at the EPIC isolator facility. The SPF mice are offspring of C57BL/6 originally acquired from the Jackson laboratory. The mice were fed a standard chow diet acquired from KLIBA NAFAG, Switzerland, ad libitum. For the low $P_i$ diet experiment, the food was changed to low phosphate chow (KLIBA NAFAG #2169) 24 h prior to infection. The mice had free access to water and were kept on a 12 h light/dark cycle at $21 \pm 1\,°C$. SPF mice were orally pre-treated with 25 mg of Streptomycin 1 day before infection, according to our established streptomycin colitis model[107]. The *S*. Tm strains used for infection were grown in LB with 3 M NaCl overnight with appropriate antibiotics. The cultures were then diluted 1:10 and subcultured in LB with 3 M NaCl for 4 h without antibiotics. The bacteria were washed in DPBS before being diluted 1:2 in DPBS and used as the inoculum ($-5 \times 10^7$ CFUs in 100 μL). To quantify CFUs in cecal tissue samples, we conducted gentamicin protection assays. For this assay, we cut the cecum open longitudinally, washed it in DPBS, and incubated it in 400 μg/ml gentamicin in DPBS (Sigma–Aldrich) at room temperature for 1 h. CFU quantification of in vivo samples was performed by plating diluted samples on MacConkey agar with appropriate antibiotics and counting. For competitive infections with two different *S*. Tm strains, the two strains were quantified by differential plating, where the two strains were quantified after plating them on solid media containing antibiotics, which only the strain of interest was resistant to. The normalized ratio is defined as the number of CFUs for the mutant divided by the number of CFUs by the isogenic parental strain, normalized to the ratio present in the inoculum. C57BL/6 mice were sacrificed by day 4 p.i. by carbon dioxide asphyxiation followed by cervical dislocation, in accordance with our license.

## Bacterial strains, plasmids, and primers used in this study
The bacterial strains used in this study are indicated in Table 1 and derived from *S*. Tm SL1344, which was passaged through a mouse and designated SB300[12,111]. Bacterial strains were cultivated in 1.5% lysogeny broth (LB) at 37 °C with shaking. LB was supplemented with ampicillin (100 μg ml⁻¹), kanamycin (50 μg ml⁻¹), streptomycin (50 μg ml⁻¹), or chloramphenicol (35 μg ml⁻¹), when necessary. All oligos and plasmids used in this study are indicated in Tables 2 and 3.

## Genetic barcoding of strains using WITS and WISH tags
WITS and WISH were introduced in the strains of interest by p22 transduction[55,56,112]. The WISH tags used in this study are a beta version of the ones published in ref. 56 that were compatible with qPCR but were only later made compatible with quantification by next-generation sequencing. All the WITS and WISH tags are integrated between *malX and -Y* genes along with a kanamycin, chloramphenicol, or ampicillin resistance cassette.

## List of fitness-neutral genetic tags
WITS01, WITS02, WITS11, WITS13, WITS17, WITS19, WITS21.
WISH028, WISH115, WISH182, WISH194, WISH205, WISH206, WISH207, WISH208, WISH209, WISH210, WISH212, WISH213, WISH214.

The sequences of WISH tags can be found in the supplementary table provided by Daniel et al., Nature Microbiology, 2024[45].

## Gene disruption
Novel gene knockouts were created using the λ/red one-step protocol, where genes were disrupted with a *cat* or *aphT* gene, conferring chloramphenicol or kanamycin resistance, respectively[113]. The primers for the knockout were designed by taking the 40 bp flanking the gene(s) of interest, followed by 20 bp for the amplification of the plasmids pKD3 (encodes the *cat* gene) and pKD4 (encodes the *aphT* gene). Primers were used to amplify the pKD3 and pKD4 plasmids and create overhangs that lead to the integration of the antibiotic resistance gene inside the gene(s) to be replaced. The PCRs were done using high-fidelity Phusion polymerase (Thermo Fisher Scientific F530L). The length of the PCR was verified on a gel and the DNA was cleaned up by ethanol precipitation. Briefly, the PCR product was pooled and precipitated in 0.1 vol of 3 M sodium acetate and 2.5 vol ethanol. The DNA was allowed to precipitate at −20 °C for 10 min before being spun down for 5 min at 4 °C and full speed. The DNA was washed twice in EtOH before being resuspended in ddH2O. *S*. Tm cells carrying the pKD46 plasmid were grown in SOB medium and induced for the expression of phage-derived genes encoded on pKD46 using

**Table 1 | Strains used in this study**

| Strain number | Strain background | Relevant genotype | Fitness neutral tag | Reference |
|---|---|---|---|---|
| T1939 | Turbo *E. coli* (DH5a) | | | New England Biolabs # C2984 |
| | SB300 | WT, SL1344 | | 12,111 |
| Z1665 | | | WITS01 (*cat*) | 121 |
| M3148 | | | WITS02 (*aphT*) | 121 |
| Z1661 | | | WISH001 | 56 |
| T3176 | | | WISH004 | 112 |
| T3289 | | | WISH010 | 112 |
| | | | WISH029 | 56 |
| T3751 | | | WISH115 | 112 |
| T3753 | | | WISH182 | 112 |
| T3755 | | | WISH194 | 112 |
| T3757 | | | WISH205 | 112 |
| T3759 | | | WISH206 | 112 |
| T3761 | | | WISH207 | 112 |
| T3763 | | | WISH208 | 112 |
| T3765 | | | WISH209 | 112 |
| T3767 | | | WISH210 | 112 |
| T3769 | | | WISH212 | 112 |
| T3771 | | | WISH213 | 112 |
| T3515, T3773 | | | WISH214 | This study |
| T256 | SB300 | *glpT::aphT* | | This study[13] |
| T258 | SB300 | Δ*glpT* | | This study |
| T2550 | | | WITS01 (*aphT*) | This study |
| T2586 | SB300 | *glpABC::cat glpD::aphT* | WISH008 | This study |
| T4583 | SB300 | Δ*glpABC* Δ*glpD* | WITS17 (*aphT*) | This study |
| T2530 | SB300 | *glpFK::cat* | WITS21 | This study |
| T2587 | SB300 | *ugpB::aphT* | WISH010 | This study |
| T2587 | | | WISH029 | This study |
| T2528 | SB300 | *mgsA::cat* | WITS17 (*aphT*) | This study |
| T4239 | SB300 | *gldA::cat gloA::aphT* | WISH004 | This study |
| T4205 | SB300 | *gloB::cat* | WISH206 | This study |
| T4206 | SB300 | *gloB::aphT* | WISH207 | This study |
| T4216 | SB300 | *gloC::cat* | WISH208 | This study |
| T4217 | | | WISH209 | This study |
| T4218 | SB300 | *gloC::aphT* | WISH210 | This study |
| T4219 | | | WISH212 | This study |
| T4240 | SB300 | *gldA::aphT gloC::cat* | WISH004 | This study |
| T3395 | SB300 | *ydhD::cat* | WISH115 | This study |
| T3396 | SB300 | *yqhD::aphT* | WISH182 | This study |
| T4203 | SB300 | *fghA::cat* | WISH194 | This study |
| T4204 | SB300 | *fghA::aphT* | WISH205 | This study |
| T4220 | SB300 | *yeaE::cat* | WISH213 | This study |
| T4221 | SB300 | *yeaE::aphT* | WISH214 | This study |
| T5110 | SB300 | *pitA::cat* | | This study |
| T5111 | | | | |
| T5116 | SB300 | Δ*glpT pitA::cat* | | This study |

**Table 1 (continued) | Strains used in this study**

| Strain number | Strain background | Relevant genotype | Fitness neutral tag | Reference |
|---|---|---|---|---|
| T5117 | | | | |
| T5174 | SB300 | *pstS::aphT* | | This study |
| T5175 | | | | |
| T5102 | SB300 | Δ*glpT pstS::aphT* | | This study |
| T5103 | | | | |
| T5106 | SB300 | *yjbB::aphT* | | This study |
| T5107 | | | | |
| T5112 | SB300 | Δ*glpT yjbB::aphT* | | This study |
| T5113 | | | | |
| T5409 | SB300 | Δ*pitA* Δ*pstS yjbB::aphT* | | This study |
| T5410 | | | | |
| T5411 | SB300 | Δ*pitA* Δ*pstS yjbB::aphT* Δ*glpT* | | This study |
| T5412 | | | | |
| T5196 | SB300 | *pitA::cat yjbB:: aphT* | | This study |
| T5197 | | | | |
| T5198 | SB300 | Δ*pitA yjbB::aphT* Δ*glpT* | | This study |
| T5199 | | | | |
| T1950 | SB300 | Δ*invG* Δ*ssaV* | WITS13 (*cat*) | This study |
| Z8081 | SB300 | Δ*invG* Δ*ssaV glpT::aphT* | | This study |
| T2500 | SB300 | *mgsA::cat* | | This study |
| T2524 | SB300 | Δ*glpT mgsA::cat* | | This study |
| T4290 | SB300 | *gloB::cat* | | This study |
| T5490 | SB300 | Δ*glpT gloB::cat* | | This study |

10 mM arabinose (Sigma–Aldrich) and then made electrocompetent. After electroporation, cells were rescued in SOC, incubated at 37 °C for 1 h, and then plated on LB with kanamycin or chloramphenicol. p-22 lysates were prepared and used to transduce the mutation into the ancestral strain and other strains of interest[114]. p-22 transduction was also used to introduce mutations available in the McClelland collection to strains of interest[115]. Where indicated, the *cat* or *aphT* resistance cassettes were removed by FLP recombination[113].

### Promoter fusion constructs
Promoter fusion constructs were created by amplifying 500 bp upstream of the gene of interest and its first three bases. The promoter region was fused to the reporter gene (gfpmut2 or a codon-optimized nanoluc) by Gibson Assembly[116]. Gibson Assembly Master Mix (New England Biolabs, E2611L) was used according to the manufacturer's instructions. Primers were designed with NEBuilder.

### Quantitative PCR (qPCR)
One hundred microliter of the homogenized samples was diluted in 2 mL of LB with Streptomycin and incubated for 4 h at 37 °C with shaking. The pellet of the enrichment culture was used for genomic DNA extraction with the QIAamp DNA Mini Kit (QIAGEN). The primers were mixed to a final concentration of 1 μM. Duplicates of the DNA samples (2 μL) were amplified using the FastStart Universal SYBR Green Master Mix (Roche, Cat# 4385610). The qPCR was carried out using a QuantStudio 7 Flex instrument (Applied Biosystems) with the following protocol: initial denaturation at 95 °C for 14 min followed by 40 cycles of 94 °C for 15 s, 61 °C for 30 s, and 72 °C for 20 s. The qPCR primers were designed to have equal amplification efficiencies[55,56]. The DNA extracted from the inoculum was used for a standard curve,

**Table 2 | Primers used in this study**

| Primer name | Primer sequence | Function | Reference |
|---|---|---|---|
| WITS01 | acgacaccactccacacct | qPCR on WITS | 55 |
| WITS02 | acccgcaataccaacaactc | qPCR on WITS | 55 |
| WITS11 | atcccacacactcgatctca | qPCR on WITS | 55 |
| WITS13 | gctaaagacacccctcactca | qPCR on WITS | 55 |
| WITS17 | tcaccagcccaccccctca | qPCR on WITS | 55 |
| WITS19 | gcactatccagccccataac | qPCR on WITS | 55 |
| WITS21 | acaaccaccgatcactctcc | qPCR on WITS | 55 |
| ydgA | GGCTGTCCGCAATGGGTC | qPCR on WITS | 55 |
| WISH028 | TTAGCGAGGTTGGCATCTTTATCC | qPCR on WISH | 56 |
| WISH031 | AAATCCATACCCGTTCCTCAATCG | qPCR on WISH | 56 |
| WISH033 | CAAGATAGTGGAGAGGTGGAATGC | qPCR on WISH | 56 |
| WISH038 | CGTGCTTATCTCTGTCTCGTTACC | qPCR on WISH | 56 |
| WISH115 | CTTTCCGTCAGAGCATACCTTACC | qPCR on WISH | 56 |
| WISH182 | CGCCTACTATGTGTCTGTGTTACC | qPCR on WISH | 56 |
| WISH194 | AATCAATACACGACGGACCTTACG | qPCR on WISH | 56 |
| WISH205 | GAACGGTAATGGCACTCTGATAGG | qPCR on WISH | 56 |
| WISH206 | GGTAGAAATGGACTGCTGGTATCG | qPCR on WISH | 56 |
| WISH207 | CAGAATGTGTAACGCCTCCTATCC | qPCR on WISH | 56 |
| WISH208 | CAGTCGGAGAAAGAAGGTCATACG | qPCR on WISH | 56 |
| WISH209 | GCTACAATCGCCTACCATCTATCG | qPCR on WISH | 56 |
| WISH210 | GCCTATCTTATCCACGCAGTATCG | qPCR on WISH | 56 |
| WISH212 | GGCTTCCATTACTCGTTTCATTGC | qPCR on WISH | 56 |
| WISH213 | TGCGAATAAACTAAGCCCGATACC | qPCR on WISH | 56 |
| WISH214 | TCCCTTTATTATGCGTCGGTTAGC | qPCR on WISH | 56 |
| WISH215 | CCAAAGCGAGAGAATCAGGTATCC | qPCR on WISH | 56 |
| WISH rev | TATGAGGAGAGTAGGAGGCAATGG | qPCR on WISH | 56 |
| ugpB_seq_F | GGACGGGATGATGACATGA | Verification of KO | This study |
| ugpB_seq_R | CCCCCGCGAACATAAAACG | | This study |
| gldA_seq_F | CGGAGAGAATGGTGCGTGAT | Verification of KO | This study |
| gldA_seq_R | GCCAGCTTTAAAACGCCGC | | This study |
| glpFK_KO_F | CGGAGTTGCCTCCGCACAAAGATTTTGCCGGATGGCGACC ata tga ata tcc tcc tta gtt | KO of glpFK | This study |
| glpFK_KO_R | CGCATACCAATAATCATTACATACTCTTCAGGATCCGATT tgt gta ggc tgg agc tgc ttc | | This study |
| glpK_seq_F | gatgtgtgtgcggagttgc | Verification of KO | This study |
| glpK_seq_R | GATACCTGCGTCGTGGAAGAG | | This study |
| glpF_seq_F | TTCATGGCGCGATAACGC | Verification of KO | This study |
| glpF_seq_R | gatgtccatgacctcctcgc | | This study |
| glpABC_KO_F | GTATGGCTAAATGATAAAAAACGAACTGTGAGGAAAAACA ata tga ata tcc tcc tta gtt | KO of glpABC | This study |
| glpABC_KO_R | AGTAGGGTAATGGCGCCGGAGGGGCTCCGGCGCTTTTTAC tgt gta ggc tgg agc tgc ttc | | This study |
| glpABC_seq_F | TGGCCGCGATGTTAAGTAAAG | Verification of KO | This study |
| glpABC_seq_F | AGTAACGTCGTTATCTGTCATGC | | This study |
| glpD_seq_F | CATTTTATCGCCCAGTGTACG | Verification of KO | This study |
| glpD_seq_R | GGAAAACTCTCTGGAATAGCG | | This study |
| mgsA_KO_F | ttccagtaatctgtagacaggttaactacggaatcgaattatatgaatatcctccttagtt | KO of mgsA | This study |
| mgsA_KO_F | cctaaagcgcgggcggtacagcatcccgcccgcgtagcgttgtgtaggctggagctgcttc | | This study |
| mgsA_seq_F | cttgatatgagttgtcccagc | Verification of KO | This study |
| mgsA_seq_R | gataatccctcgcgctttgtg | | This study |
| pitA_ver_F | CTCAAAATGGCGTAACGTCC | Verification of KO | This study |
| pitA_ver_R | GATAAGCCTCCGGATGACG | | This study |
| GA_PglpT_F | cgtaatacgactcactatagCGGTAACCGCATAACGAG | Gibson Assemblyof pZ8001 | This study |
| GA_PglpT_R | aatactacctcctgaattccATTGTAGCCTCCGTGGCC | | This study |
| pM965_amp_F | GGATCCCCCGGGCTGCAGGAATTC | | This study |
| pM965_amp_R | AGCGGCCGCCACCGCGGT | | This study |
| pZ8049_PpstS_F | cacctaaaATGGCGGCAGGGATGGTAC | Gibson Assembly of pZ8049 | This study |
| pZ8049_PpstS_R | acaccataCATAATGTCTCCTGCACGGTTTC | | This study |
| pZ8049_pZ_F | gagacattatgTATGGTGTTCACACTGGAAG | | This study |

**Table 2 (continued) | Primers used in this study**

| Primer name | Primer sequence | Function | Reference |
|---|---|---|---|
| pZ8049_pZ_R | gccgccatTTTAGGTGGCACTTTTCGG | | This study |
| pZ8040_ F | gccacctaaaTATGGTGTTCACACTGGAAG | Gibson Assembly of pZ8040 | This study |
| pZ8040_ R | gaacaccataTTTAGGTGGCACTTTTCG | | This study |
| pZ8046_PgloC_F | cacctaaaAGCTGCTGGCGCTTGGTG | Gibson Assembly of pZ8046 | This study |
| pZ8046_PgloC_R | acaccataCATACTGCTCCTTTGATTAAGCCTC | | This study |
| pZ8046_BB_F | gagcagtatgTATGGTGTTCACACTGGAAG | | This study |
| pZ8046_BB_R | ccagcagctTTTAGGTGGCACTTTTCGG | | This study |
| pZ8044_rpsM_F | cacctaaaTCTAGACGATAAAGTAATGACCCG | Gibson Assembly of pZ8044 | This study |
| pZ8044_rpsM_R | acaccataCGGGCCACTATGCACTCC | | This study |
| pZ8044_BB_F | tggcccgTATGGTGTTCACACTGGAAG | | This study |
| pZ8044_BB_R | atcgtctagaTTTAGGTGGCACTTTTCGG | | This study |
| pZ8026_glpTcomp_F | cgtaatacgactcactatagCGGTAACCGCATAACGAG | Gibson Assembly of pZ8026 | This study |
| pZ8026_ glpTcomp _R | aatactacctcctgaattccTTAGCCTCCGTTGCGTTTTAAC | | This study |
| pZ8026_BB_F | GGAATTCAGGAGGTAGTATTG | | This study |
| pZ8026_BB_R | CTATAGTGAGTCGTATTACGC | | This study |

**Table 3 | Plasmids used in this study**

| Plasmid name | Relevant genotype | Antibiotic resistance used for selection | Ori / backbone | Reference |
|---|---|---|---|---|
| pKD3 | FRT-cat-FRT | Cm | | 113 |
| pKD4 | FRT-aphT-FRT | Kan | | 113 |
| pKD46 | mScarlet counter-selection cassette | Amp | | 113 |
| pCP20 | FLP recombinase | Amp, Cm | | 113 |
| pZ8001 | $P_{glpT}$-gfpmut2 | Amp | pWKS30/pSC101 | This study |
| pM3101 | promoterless-gfpmut2 | Amp | pSC101 | 121 |
| pZ8049 | $P_{pstS}$-nanoluc | Amp | pSC101 | This study |
| pZ8046 | $P_{gloC}$-nanoluc | Amp | pSC101 | This study |
| pZ8040 | promoterless-nanoluc | Amp | pSC101 | This study |
| pZ8044 | $P_{rpsM}$-nanoluc | Amp | pSC101 | This study |
| pZ8026 | $P_{nat}$-$glpT$ | Amp | pSC101 | This study |
| pZ7903 | Empty vector | Amp | pSC101 | Kindly provided by Thea Bill Andersen (manuscript in preparation) |

which was used to calculate the CT values and the relative frequency of every barcode in the pooled genomic DNA.

## Defined media for $P_i$ titration and the $P_{pstS}$-nanoluc reporter assay

Bacteria were grown in a minimal medium: 0.5 g/L NaCl, 1 g/L NH4Cl, 100 mM CaCl2, 2 mM MgSO4 supplemented with 250 mg/L Histidine-HCl, 4 mM fumarate and 40 mM pyruvate. For the overnight growth, 1 mM $P_i$ in the form of K2HPO4 was added to the medium. In the morning, cells were diluted 1:100 in the same medium. After 4 h, cells were washed and resuspended in the base medium free of added $P_i$ with a titration of K2HPO4 at the indicated concentration. The time of resuspension in the medium with the addition of the indicated $P_i$ concentration is considered time point 0 h. The cultures were incubated at 37 °C in aerobic conditions and with shaking. After 24 h, the $OD_{600-nm}$ was measured, the cells were pelleted and frozen until the luciferase assay was conducted. The reporter activity was normalized to the $OD_{600 \, nm}$.

## Luciferase assay

A nanoluciferase gene was codon-optimized and synthesized by IDT technologies. In vivo, samples were centrifuged at $30 \times g$ for

1 min to remove debris. All samples were centrifuged at $21,000 \times g$ for 5 min. The cells in the pellet were lysed by exposing them to 3 cycles of cold shock in liquid nitrogen followed by 5 min at 98 °C. The lysed cells were resuspended in the Promega NanoGlo luciferase buffer and processed according to the vendor's protocol.

## Antibody-mediated depletion of macrophages and neutrophils

Anti-CSF1R (LuBio Science #BE0213) antibodies were used for macrophage depletion. The antibodies were diluted in DPBS to a concentration of 5 g/L for the first dose and 1.5 g/L for injections on days 0, 1, 2, and 3 p.i. 200 μl of antibody solution was administered with every injection using a 1 mL syringe and a 26 G needle. To control for the effect of the injection, the cohort of control mice was injected with 200 μl of DPBS or 150 μl of isogenic antibody control (IgG2a raised in rats, LuBio Science #BE0089).

## Total inorganic orthophosphate quantification

Samples were collected in water instead of DPBS, and phosphate levels were quantified using a commercially available kit (Sigma–Aldrich MAK030) according to the manufacturer's instructions.

## Bacterial flow cytometric analysis

Fecal and cecal content samples were harvested in 1 mL PBS and homogenized for 2.5 min at 25 Hz. The samples were then centrifuged for 1 min at $500 \times g$ and RT to remove debris. The supernatant was transferred to a 1.5 mL Eppendorf tube, which was centrifuged for 5 min at $1500 \times g$ and 4 °C. The pellet was resuspended in 300 µl of DPBS and transferred to a 96 well plate. The plate was centrifuged at $1500 \times g$ and 4 °C for 5 min. The pellet was resuspended in 200 µl of 2% PFA in DPBS for fixation and incubated for 20 min at RT. Then the cells were washed in 200 µl of 1% BSA in DPBS. Next, the cells were resuspended in 50 µl of 1% BSA in DPBS containing human anti-O12 antibody (hSTA5, 1:200). The samples were incubated with the primary antibody for 30 min at RT and in the dark. Then 150 µl of 1% BSA in DPBS were added and the samples were washed with 200 µl of 1% BSA in DPBS. The pellet of bacterial cells was resuspended in 50 µl of 1% BSA in DPBS containing anti-human IgG Fc AF647 (1:200) and incubated for 30 min at 4 °C and in the dark. 150 µl of DPBS were added and the samples were washed with 200 µl of DPBS. The cells were resuspended in 200 µl of DPBS and analyzed using a CytoflexS cytometer (Beckmann Coulter).

## Targeted liquid chromatography mass spectrometry

Murine cecal contents were sampled for the quantification of short-chain fatty acids. After harvesting, the samples were immediately derivatized with 3- Nitrophenylhydrazine and N-(3-dimethylaminopropyl)-N'-ethylcarbodiimide hydrochloride in 70% isopropanol in LCMS-grade $H_2O$. Short-chain fatty acids analysis by reversed-phase LC-MS was carried out on an Ultimate 3000 ULPC instrument (Thermo Scientific, Waltham, MA, USA) hyphenated to a QExative plus mass spectrometer (Thermo Scientific, Waltham, MA, USA) as described in[117] with slight modifications. For LC separation we used a Kinetex XB C18 column (particle size 1.7 µm; $50 \times 2.1$ mm, Phenomenex, Torrance, CA, USA) with 0.1% formic acid in water (solvent A) and 0.1% formic acid in acetonitrile as mobile phases. The sample injection volume was 2 µl. We ran a linear gradient at a flow rate of 500 µl min$^{-1}$ with solvent B changing as follows: 0 min: 2%; 3 min: 95%; 5 min: 95%; 5.3 min:2%. Subsequently, the column was equilibrated for 2 min at initial condition. Mass analysis was performed in the negative FTMS mode at a mass resolution of 70,000 (at m/z 200) by applying heated electrospray ionization. Source parameters were set as follows: spray voltage: 2.7 kV; S-lens RF level: 50; sheath gas: 50; aux gas: 20; sweep gas: 0; heater: 350 °C. The metabolites were quantified using stable isotope-labeled as internal standards [$^{13}C_4$]-fumaric acid (99%), and [$^{13}C_4$, $^{15}N$]-L-aspartic acid (99%), purchased from Cambridge Isotope Laboratories.

## Histopathology of in vivo samples

Murine cecal tissue sections were carefully dissected, embedded in Optimal Cutting Temperature embedding medium (O.C.T., Tissue Tek), and flash-frozen in liquid nitrogen. Samples were stored at −80 °C until they were processed. Samples were cut in 5 µm sections using a CryoStar NX50 Cryostat (Fisher Scientific) and air-dried on glass slides. The samples were stained with hematoxylin and eosin, and images were acquired on an AxioVision light microscope.

## Phylogenetic analysis

An enzyme similarity analysis of the SL1344 GlpT (UniProtKB #A0A0H3NJJ9) amino acid sequence was conducted with EFI, where the UniProt database (version 2024–01) served as the reference [118,119]. The search retrieved 792 similar proteins within the Enterobacteriaceae family. All subsequent analyses were performed using MEGA7[120]. The Sequence identity matrix was calculated with the Poisson model and visualized as a matrix of the within and between genus identity. All sequences are accessible on the public UniProt database.

## Statistical analysis

All statistical analyses were conducted using GraphPad Prism 9 or 10 for Windows. Where applicable, a paired or unpaired $t$-test, Wilcoxon test, One-way or Two-way ANOVA test were used to assess statistical significance. The test used for each graph is indicated in the figure legends.

## Biological materials availability

Those interested in our strains, plasmids, and/or tags are encouraged to contact Prof. Wolf-Dietrich Hardt (wolf-dietrich.hardt@micro.biol.ethz.ch).

## Reporting summary

Further information on research design is available in the Nature Portfolio Reporting Summary linked to this article.

# Data availability

All data needed to evaluate the conclusions of this study are presented in the Article and Supplementary Information. Additional raw data, such as flow cytometry raw data, microscopy metadata, and mass spectrometry raw data relevant to this paper, will be made openly accessible by the ETH Research Collection in agreement with ETH Zürich's open-access policy. Source data are provided with this paper.

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

## Acknowledgements

We want to acknowledge the highly appreciated work of the staff at the ETH animal facility (EPIC and RCHCI, especially Manuela Graf, Katharina Holzinger, Dennis Mollenhauer, Sven Nowok, and Dominik Bacovcin). We are grateful to Patrick Andreassen, Joshua Newson, and Luca Maurer for the introduction to macrophage infections, intraperitoneal injections, and the microtome, respectively. We also want to thank Dirk Bumann, Randall Platt, and members of the Hardt, Sunagawa, and Slack labs for helpful comments and discussions. Furthermore, we appreciate Andrew Abi Younes and Patrick Andreassen's input and feedback on this manuscript. NSdS was funded by a Boehringer Ingelheim Fonds PhD fellowship. The Swiss National Science Foundation funded this work via grant 310030_192567 to WDH and via the NCCR Microbiomes grant to JAV and WDH. C.S is supported by the German Research Foundation (SCHU 3606/1-1). Y.C. is supported by an EMBO long-term fellowship (ALTF-234-2020) and a flexibility grant from the SNF/NCCR Microbiome (51NF40_180575). Figures 1a, 2d, 3a, 4a, b, and 5b were created by the first author (NSdS) for this study using BioRender.com. The figures are subject to a CC-BY license, where we hold publication rights granted by BioRender.com.

## Author contributions

NSdS, BDN, CS, and WDH conceived the project and designed the experiments. NS carried out the mouse experiments and analyzed the data. NSdS, MV, TBA, EB, CS, and BDN cloned the strains, plasmids, and tags. NSdS and SK carried out flow cytometry experiments. MB performed histopathology. PC, PK, and JAV contributed the LC-MS measurement and analysis. WDH provided the resources for the study. NS wrote the initial draft of the manuscript. NSdS, YC, EB, and WDH edited the draft. All authors read, commented on, and approved this manuscript.

## Competing interests

The authors declare no competing interests.
