## [Transparent Peer Review file · Nature Communications]

Context-dependent change in the fitness effect of (in)organic phosphate antiporter *glpT* during *Salmonella* Typhimurium infection

Corresponding Author: Professor Wolf-Dietrich Hardt

Version 0:

Reviewer comments:

Reviewer #1

(Remarks to the Author)

In the manuscript entitled “The context-dependent change in the fitness effect of the (in)organic phosphate antiporter *glpT* during *Salmonella* Typhimurium infection” Souza et al. investigate the underlying costs and benefits associated with diversification of the *glpT* locus within *Salmonella enterica* serovar Typhimurium (Stm) populations, as it transitions through the mice gut and crosses into lamina propria to begin the invasion of deeper tissues. The authors find that, at the beginning of infection, mutations in *glpT* that inactivate the resulting protein expand in the bacterial population because the expression of GlpT in the gut is costly. The authors provide evidence suggesting that GlpT-expressing bacteria may be outcompeted by *glpT* mutants because GlpT inhibits bacterial growth by over-importing phosphate (Pi) that is abundantly present in the gut lumen. As the infection progresses, Stm promotes gut inflammation via its type 3 secretion systems (T3SSs), leading to the recruitment of macrophages that try to control Stm cells that are replicating in the gut lumen and lamina propria. At this stage of the infection, the authors provide evidence suggesting that *glpT* null mutants have a selective disadvantage because GlpT appears to promote the expression of enzymes in the methylglyoxal detoxification pathway that is required for intramacrophage replication. The work is exciting as it would provide a rationale for why the frequency of specific alleles changes in the bacterial population as it transitions through its ecological niches. Nonetheless, some of the ideas could be better articulated, and the proposed model could be bolstered or undermined by additional experiments.

Please find specific comments below:

1. Sentence in lines 72-73 is unclear.
2. Line 85, the meaning of the term “evolutionary conflict” is unclear.
3. Lines 102-113:
 - a. In Fig. 1, please include the level of polymorphism resulting in null mutations in a typical, selectively neutral site (e.g. a gene that has been shown to be dispensable for replication in the gut lumen and macrophages)? Perhaps a graph could be included, displaying the average null mutation rate across the genome and a spike in enriched regions such as the *glpT* ORF. If selectively neutral genes also show elevated polymorphism, one of them should have been used as a control in these initial experiments.
 - b. Please state whether null mutations in *glpT* are also detected in mice harboring microbiota (from Ref. 13?).
 - c. Does a *glpT* null mutation attenuate Stm virulence in a single strain infection? Do mice show increased resistance to *glpT* null mutants? Is there a delay on mice killing when they are infected with *glpT* null mutants? If not, then it would be valuable if the authors could elaborate, in the discussion, how selection during host interactions could partition the bacterial population, promoting and homogenizing genetic diversity within subpopulations.
4. Line 122, YjbB has been proposed to be a Pi exporter (PMID: 2148893). Why was this gene deleted here?

5. Lines 140-158:

a. In Fig. 3b, how were these experiments conducted? How were the bacteria grown? What medium was used? What temperature? Were cultures aerated with agitation? When were measurements conducted? I presume that measurements were not performed in lysogeny broth (LB) as described in the methods because LB has Pi concentrations in the mM range. Do all strains have similar growth rates? Are measurements being performed in the same growth phase? How would growth rate affect reporter activity? Why was nanluc used instead of a kinetic reporter?

b. Genes with apparently redundant functions can play roles in specific ecological contexts (for instance PMID: 29259085, PMID: 30963998 or PMID: 31753999). *Stm* has two additional Pi transporters (*PitA* and *Pst*), which, as far as we know, are more efficient in importing Pi than *GlpT* (PMID: 3522583, PMID: 6998957; PMID: 11489853). While it is feasible that *GlpT* is more efficient in certain environments, like the gut, the authors have not made this point.

c. *GlpT* has been shown to import Pi (PMID: 3522583) and balance intracellular Pi levels in the bacterial cytosol (i.e. it exports intracellular Pi when glycerol-3-phosphate is over-imported via the *Ugp* system (e.g. see ref 26)). Rather than repeating the former point, the authors need to emphasize that this protein can promote the accumulation of Pi into the bacterial cytoplasm through non-equimolar Pi exchanged (PMID: 15556940 and ref 28).

d. Reference 41 reports that *pstS* transcription is active intracellularly, when *Stm* is replicating inside a macrophage, not that *pstS* is a reporter for intracellular Pi in the bacterium. The inference that *pstS* is transcribed in response to a decrease in intracellular Pi has been proposed by other studies (PMID: 36993483; PMID: 2943772).

6. Lines 160-178, the inference from indirect data using *Stm* mutants of known pathways does not convince this reviewer that glycerol/glycerol-3-Pi metabolism is not active during luminal growth. Glycerol-3-Pi can be generated by *GpsA*, which is not considered in these experiments. Additionally, other unknown enzymes could promote glycerol/glycerol-3-Pi metabolism in this context. If bacteria were to accumulate glycerol-3-Pi during luminal growth, the presence of *GlpT* could reduce fitness by promoting the exchange of this metabolite with Pi present in the gut lumen. What happens if mice are fed a diet rich in glycerol-3-phosphate? Does the *glpT* locus become selectively neutral? Does the wild-type strain become fitter than the *glpT* mutant?

7. The sentence in lines 277-231 is unclear. If the host restricts Pi availability, a mutation in other transporters should impact intracellular growth. Accordingly, a deletion of the *Pst* transporter has been shown to lower *Stm* fitness in macrophages (Ref. 41). Based on this, we can infer that the fitness of the $\Delta 3\text{Pi}$ strain would be even lower.

8. Lines 239-241, the increase in *gloC* promoter activity is consistent with the proposed role of *GlpT* in methylglyoxal detoxification. However, what happens to the fitness/replication of the *glpT* vs wild-type strain in a background that is impaired for methylglyoxal detoxification (e.g. *mgsA*, *gloB* or *gloC*)? Does the presence of *GlpT* become irrelevant in a *mgsA*, *gloB* or *gloC* background (ref. 51)?

9. Lines 292-303:

a. If the *GlpT* decreases fitness in the lumen by importing excess of Pi into the cytosol of *Stm*, would over-expression of the Pi exporter *YibB* (PMID: 2148893) rescue this phenotype?

b. The really puzzling fact about the data is that given that *glpT* can be regulated (https://bioinf.gen.tcd.ie/cgi-bin/salcom.pl?header_rotation=45;query=glpT;db=salcom_mac_HL), why is this protein being expressed in the lumen (or is it)? Does *Stm* inhibit *GlpT* expression, but the small amount of *GlpT* that is being produced is detrimental? This phenomenon is reminiscent of other studies that have shown that *Escherichia coli* accumulates mutations in regulated proteins as it passes through the mammalian gut (PMID: 21698140).

Reviewer #2

(Remarks to the Author)

Based on the reasonable working hypothesis a part of which is supported by their previous report, the author assessed the fitness effects of *glpT* mutations in *S. enterica* during multiple phases of orogastric mouse infection. The authors found that *glpT*-deficient mutants outcompete WT at initial growth in the gut lumen when phosphate availability is high. They also showed that the ratio of *glpT*-deficient mutants and WT changed at the later phase depending on macrophages. Their findings may be interesting, but some conclusions are not clearly evident and/or explained. Also, the writing in the manuscript needs to be improved.

Major comments:

(1)

One of the keys of this work is that the author found a metabolic trade off for the fitness. Because other trade-off genes such as *T3SS* and *flagellin* are known, the author should describe more how the discovery of a metabolic gene is significant, probably in the introduction.

(2)

The author showed that the fitness of *glpT*-deficient mutants in *S. enterica* changed over time (day 0-day 4), and also

mentioned that this change is due to the change of environment from “in the gut lumen” to “inside phagocytes”. The reviewer thinks that the author showed that this change is dependent on the macrophage, but did not demonstrate that *S. enterica* cells are actually present in the macrophages.

(3) L116-L129

It is not easy to understand how the results shown in Fig.2a, b, d can be explained in a consistent manner. The reviewer suggests that the authors clearly explain possible interpretation(s) for these results step by step. This may help readers for subsequent reading.

(4)

It seems that the manuscript has not yet been finalized: Table 2 is incomplete, Fig.S2c and S4d are not cited (might be due to mistakes). Some parts are not easy to understand because important information seems to be missing: detailed comments are provided below.

Other comments:

(5) L107 “differential plating”

The reviewer thinks that the authors need to describe clearly and with more details how this was performed to show that the obtained numbers or the calculated ratios are reliable. It is desired to exhibit some raw data examples.

(6) L109 “Notably, the WT outcompeted from day 2 and the mutant/WT ratio progressively declined 110 from around 10 to around 1 by day 4 p.i. (Fig. 1b).”

Is this tendency conserved within individual mice? Or some mice showed different patterns?

The reviewer suggests that the multiple dots at different timepoints from same mouse are linked.

(7) L122

Current Fig. 2c should be cited here and labeled as Fig. 2b.

(8) L143

It is not clear how the three scenarios are selected to show.

(why other scenarios are not illustrated?)

Also, the meaning of illustrations are not clear:

e.g., why does not scenario 3 show Gly-3-P?

why is the number of Gly-3-P and Pi different between scenario 1 and 2?

(9)

L147

Sometimes “mutant” is not clear.

Can the authors use, e.g., “knock out” or something related?

(10)

L153 “final”

What does the “final” mean?

If this is a time course measurement, the authors need to define the time “0 h”.

(11)

L155

The reviewer suggests that the author specifically mention what the “environmental context” are. Also, how is this statement related to the main conclusion in this study?

(12)

L156 “The reduction in the fitness advantage...”

Is this the authors experimental results?

If so, please site a figure or something which supports this statement.

(13)

L165 “Since the *ugp* operon is induced during phosphate limitation,”

Is this reported result? If so, please site a reference.

(14)

L166 “the ratio”

What is “the ratio” specifically?

(15)

L167 “The ratio for the *glpFK* mutant suggested that glycerol uptake and phosphorylation do not influence the fitness in the lumen”

This sentence is too sudden. For example, the author should explain what *glpFK* is in the text.

(16)

e.g., Line 266-272

The reviewer suggests that some results or direct interpretations mentioned here can be described in the result section right after the corresponding experimental results, so that readers can understand and interpreted the results easier.

(17)

L291

In terms of equilibrium, ATP concentration may be increased because of more Pi. Please check it.

Version 1:

Reviewer comments:

Reviewer #1

(Remarks to the Author)

The authors have addressed all my concerns. They I would like to congratulate them for the outstanding work.

Reviewer #2

(Remarks to the Author)

The review thinks that the authors' explanation and modifications satisfied the reviewers' comments.

Point by point response to the reviewers' comments

Reviewer #1 (Remarks to the Author):

In the manuscript entitled “The context-dependent change in the fitness effect of the (in)organic phosphate antiporter *glpT* during *Salmonella Typhimurium* infection” Souza et al. investigate the underlying costs and benefits associated with diversification of the *glpT* locus within *Salmonella enterica* serovar *Typhimurium* (*Stm*) populations, as it transitions through the mice gut and crosses into lamina propria to begin the invasion of deeper tissues. The authors find that, at the beginning of infection, mutations in *glpT* that inactivate the resulting protein expand in the bacterial population because the expression of *GlpT* in the gut is costly. The authors provide evidence suggesting that *GlpT*-expressing bacteria may be outcompeted by *glpT* mutants because *GlpT* inhibits bacterial growth by over-importing phosphate (Pi) that is abundantly present in the gut lumen. As the infection progresses, *Stm* promotes gut inflammation via its type 3 secretion systems (T3SSs), leading to the recruitment of macrophages that try to control *Stm* cells that are replicating in the gut lumen and lamina propria. At this stage of the infection, the authors provide evidence suggesting that *glpT* null mutants have a selective disadvantage because *GlpT* appears to promote the expression of enzymes in the methylglyoxal detoxification pathway that is required for intramacrophage replication. The work is exciting as it would provide a rationale for why the frequency of specify alleles changes in the bacterial population as it transitions through its ecological niches. Nonetheless, some of the ideas could be better articulated, and the proposed model could bolster or undermined by additional experiments.

Response: Dear reviewer 1, thank you very much for your interesting and insightful comments and questions! We highly appreciate your help to further improve the quality of our paper. We tried our best to address the points you have raised as described in our responses, below and have revised our manuscript, accordingly.

All the best,
Wolf & Noemi, and the rest of the team

1. Sentence in lines 72-73 is unclear.

Thanks for pointing that out, we have simplified the explanation of why an enrichment of premature stop codons serves as evidence for opposing selective pressures acting on a gene. Please find the changes in the text highlighted in red below.

Line 83 The fitness conferred by a particular gene is determined by the context of the selective pressures, meaning that the inactivation of some genes is selected in some niches, while the functional gene is selected for in others. A signature of these types of genes conferring context-dependent fitness is the enrichment for loss-of-function mutations such as premature stop codons, compared to genes that are important for fitness in all niches. ~~A hallmark of genes with highly context-dependent fitness is the enrichment of premature stop codons, which usually result in their loss of function. Such an enrichment suggests that despite being conserved, these genes experience selection-driven inactivation in certain conditions.~~ A recent screen of over 100'000 *S. Tm* genomes found that the flagellin methylase *fliB*,

which contributes to adhesion and host cell invasion, has four times as many premature stop codons **as would be expected under neutral selection**^{1,2}. These mutations are likely to have occurred/been selected in natural environments rather than inside laboratories^{2,19,20}. **The increased frequency of premature stop codons, suggests that strains proficient for the genes are favored in certain conditions, whereas other niches select for inactivating mutations**².

2. Line 85, the meaning of the term “evolutionary conflict” is unclear.

Response: Thank you for raising the term's unclarity. We initially used this term to point out that there seems to be a selective pressure selecting for the maintenance of *glpT* (high conservation within Enterobacteriaceae, see Fig. S1), as well as a selective pressure against *glpT* (evidence for selection for inactivating mutations in^{2,3}). As a result, the gene faces opposing selective pressures, resulting in multiple conflicting genetic variants or antagonistic pleiotropy.

To clarify the meaning, we replaced the term “evolutionary conflict” with “opposing selective pressures acting for/against functional *glpT*.”

3. Lines 102-113:

a. In Fig. 1, please include the level of polymorph resulting in null mutations in a typical, selectively neutral site (e.g. a gene that has been shown to be dispensable for replication in the gut lumen and macrophages)? Perhaps a graph could be included, displaying the average null mutation rate across the genome and a spike in enriched regions such as the *glpT* ORF. If selectively neutral genes also show elevated polymorphism, one of them should have been used as a control in these initial experiments.

Response: Thank you for raising that important point. We agree that indicating the average null mutation rate substantially helps with interpreting the presented data. The data generated by Cherry, GBE, 2020, takes neutral mutation rates into account and only reported premature stop codons at increased frequencies. In the long-term infection experiments performed by Gül et al, 2023, mutations were identified in 44 different genes (including *glpT*) for the clones isolated from day 157 to 160 post-infection. Previous work has also hinted at selection for mutants in many of the other genes that harbored mutations in the Gül et al., 2023 study. To illustrate which genes were affected by different independent mutations and therefore show evidence of strong selection for inactivating mutations, we created a plot indicating the number of variants per mutated gene, the frequency of clones that harbored mutations, and the start position of the gene (see below). The graph is similar to one published recently by Grote et al in Cell Host & Microbe for a study following *S. enterica*'s evolution during chronic infection⁴. We included the graph in Fig. S1 in the manuscript and described it in the text. This provides additional evidence for the strong positive selection of *glpT* mutants.

Changes in the manuscript:

Line 88 A **recent** screen of over 100'000 *S. Tm* genomes found that the flagellin methylase *fliB*, which contributes to adhesion and host cell invasion, has four times as many premature stop codons **as would be expected under neutral selection**^{1,2}.

Line 97 Recently, our lab studied within-host evolution using an *S. Tm* long-term murine infection model³. Strikingly, *glpT* mutations were found in 70 % of isolates, including single nucleotide polymorphisms (SNPs) and deletions (“dels”) at distinct locations (illustrated in **Fig. 1a**)³. Moreover, the gene is not encoded in a genomic region that displays an unusually high frequency of mutations. Together, this indicates that these mutants were positively selected within the host (**Fig. S1a**).

b. Please state whether null mutations in *glpT* are also detected in mice harboring microbiota (from Ref. 13?).

Response: Thanks for asking. In Gül et al (PLoS Biol 2023; Ref 13 in our original manuscript)³, the central

Figure S1 – Most commonly mutated genes and conservation of GlpT in Enterobacteriaceae

a The number of distinct variants per gene for all genes that harbored mutations after 157 – 160 days of infection in Gül et al., 2023. The plot shows the gene start position in the genome and indicates that *glpT* is not part of a genomic region with an unusually high mutation frequency. Selected genes that have previously been identified to be subject to positive selection for their loss of function are indicated: *fliC* and *fliB*, required for motility^{2,66,67}; *oafA*, which encodes the O5-antigen^{68,69}; *hilD*^{2,3,34}; *melR*, a positive regulator of the melibiose operon²; *barA*, which was recently shown to mutate in clinical *S. enterica* isolates⁴; and *tsr*^{2,70}.

b The frequency of isolated clones that harbored mutations in individual genes after 157 – 160 days of infection in Gül et al., 2023, and their gene start position in the genome. The *oafA* and *glpT* open-reading

experiment was performed in mice which were pre-treated with the antibiotic streptomycin on the day before infection. This pretreatment transiently disrupts colonization resistance by microbiota for 1-3 days. However, not all the gut microbiota are killed in the process, and unless *S. Tm* can suppress microbiota regrowth by causing inflammation, the *S. Tm* loads decline over time⁵. For details, please refer to papers such as⁵⁻⁷, for a characterization of the streptomycin-pretreatment mouse model and, e.g., differences to germ-free mice that do not harbor a significant microbiota. Since the experiments in the Gül et al paper ran for quite some time (157-160 days) and the microbiota disruption by streptomycin is transient, (part of) the microbiota regrew. The microbiome diversity, measured using the Shannon Index on day 160 p.i., indicates a reduced diversity (1 vs 3 for the unperturbed control³) when the animals were infected with wildtype *S. Tm* alone. In contrast, a second group of mice was co-housed after day 70 with mice that harbored an unperturbed native microbiota and showed the same within-sample diversity as the unperturbed control³. Interestingly, in these

mice Gül et al also isolated a *glpT* mutant³. As the mice were only sampled after co-housing we cannot say when this mutant arose and how it was selected.

It is worth noting that we found the luminal *glpT* phenotype to be present in competitive infections of mice with microbiota of multiple compositions: streptomycin-pretreated 129SvEv mice, long-term model³, C57BL/6 low complexity microbiota (which were infected without antibiotic pre-treatment)⁸, and streptomycin-pretreated C57BL/6 (this study).

We have further specified the conditions in which the mutations were detected in the manuscript. More specifically we added the following text highlighted in red:

Line 98 Strikingly, *glpT* mutations were found in 70 % of isolates, including single nucleotide polymorphisms (SNPs) and deletions (“dels”) at distinct locations (illustrated in **Fig. 1a**)³. **This included cases where wildtype *S. Tm* prevented the re-establishment of a complex gut microbiota and a case where the microbiota was re-established by co-housing³.**

c. Does a *glpT* null mutation attenuate *Stm* virulence in a single strain infection? Do mice show increased resistance to *glpT* null mutants? Is there a delay on mice killing when they are infected with *glpT* null mutants? If not, then it would be valuable if the authors could elaborate, in the discussion, how selection during host interactions could partition the bacterial population, promoting and homogenizing genetic diversity within subpopulations.

Response: This is a very interesting question! Swiss animal law prohibits extending the duration of our infections to death. So, running such an experiment is not an option. However, colleagues of ours have conducted competitive infections using a typhoid model, where mice are infected intravenously such that the intestinal lumen colonization is bypassed. They showed that a *glpFK glpT* mutant has a disadvantage compared to its isogenic wildtype in the murine spleen, a site where *S. Tm* is thought to depend on intracellular replication^{9,10}. In our experience the *S. Tm* burden at systemic sites correlates with the overall disease burden of mice. Given the disadvantage of the *glpFK glpT* mutant at systemic sites, it is highly likely, that there would be a delay in mouse killing for a single strain infection, if that was a possible readout. Please refer to lines 324-326 in the original manuscript, or lines 425-427 of the revised manuscript:

*“Furthermore, G3P metabolism has been found important for *S. Tm* to colonize the spleen of mice, a site where the majority of *S. Tm* reside in red pulp macrophages^{9,10}.”*

4. Line 122, *YjbB* has been proposed to be a *Pi* exporter (PMID: 2148893). Why was this gene deleted here?

Response: That is a good point. We considered using an *yjbB*-proficient strain but decided to take the more conservative approach of knocking it out, since we could not rule out that *YjbB* might also import *Pi*, in the absence of the *Pst* complex and *PitA* (as was reasoned by our colleagues in¹¹). We added the following comment in the manuscript describing our rationale:

Line 156 **“While *YjbB* has been shown to be able export *Pi* when overexpressed, we did not know its transport activity in *S. Tm* at normal expression levels and in the absence of *pitA* and *pstS*^{12,13}. Thus, we**

decided to take the conservative approach of removing it too, which should, if anything, reduce the effect of potential GlpT P_i transport, if YjbB acts as a P_i exporter in the tested conditions ¹¹.”

A study published after our first submission showed that expression of *yjbB* from a plasmid leads to an increase in P_{pstS} activation in *S. Tm* ¹³, indicating that YjbB also acts as a P_i exporter in *S. Tm* in addition to YjbB in *E. coli* ¹². If YjbB would lead to P_i export in our in vitro experiments, it should lower the effect of GlpT on P_{pstS} activation and growth in a phosphate-limited medium, making our finding of GlpT's impact on intracellular P_i levels in the absence of YjbB more significant.

5. Lines 140-158:

a. In Fig. 3b, how were these experiments conducted? How were the bacteria grown? What medium was used? What temperature? Were cultures aerated with agitation? When were measurements conducted? I presume that measurements were not performed in lysogeny broth (LB) as described in the methods because LB has Pi concentrations in the mM range. Do all strains have similar growth rates? Are measurements being performed in the same growth phase? How would growth rate affect reporter activity? Why was nanluc used instead of a kinetic reporter?

Response: Thank you for pointing out that additional information on the experimental procedure for this assay is important! We have provided additional information in the materials and methods section and in the captions for Figures 3.b and 3.c. All strains carried the same reporter plasmid.

As shown in **Fig. 3c**, the $\Delta pitA yjbB::aphT \Delta glpT$ mutant strain reached a lower OD_{600nm} than the $\Delta pitA yjbB::aphT$ mutant after 24 h of incubation. We accounted for that difference in maximal OD by dividing the reporter activity (arbitrary units of luminescence) by the OD_{600nm} (**Fig. 3b**), as is commonly done. The normalized luminescence (lum/OD) of the constitutive expression control does not change significantly (**Fig. 3a**). Bruna et al., 2024 report that activation of their P_{pstS} reporter was independent of the growth rate. Since our measurements were taken after 24 h of growth, we expect all cultures to be in the stationary phase.

We added the following paragraph to the Material and Methods section:

“Defined media for P_i titration and the P_{pstS}-nanoluc reporter assay

Bacteria were grown in a minimal medium: 0.5 g/L NaCl, 1 g/L NH₄Cl, 100 mM CaCl₂, 2 mM MgSO₄ supplemented with 250 mg /L Histidine-HCl, 4 mM fumarate and 40 mM pyruvate. For the overnight growth, 1 mM P_i in the form of K₂HPO₄ was added to the medium. In the morning, cells were diluted 1:100 in the same medium. After 4 h, cells were washed and resuspended in the base medium free of added P_i with a titration of K₂HPO₄ at the indicated concentration. The time of resuspension in the medium with the addition of the indicated P_i concentration is considered timepoint 0 h. The cultures were incubated at 37 °C in aerobic conditions and with shaking. After 24 h, the OD_{600nm} was measured, the cells were pelleted and frozen, until the luciferase assay was conducted. The reporter activity was normalized to the OD_{600nm}.”

And specified when the measurement was taken in the figure legend:

Figure 3 GlpT can act as a phosphate importer

b P_{pstS} promoter activity **after 24 h** as a readout for intracellular phosphate levels reveals that GlpT can increase intracellular phosphate levels. The statistical significance was calculated with an ordinary Two-Way ANOVA with Šidák multiple test correction to account for the effect of the genotype and concentration. Statistical significance was tested using an ordinary one-way ANOVA. The lines represent the mean, while the whiskers range from the smallest to the largest value. $n = 3$ biological replicates.

On using nanoluc rather than a kinetic reporter:

Our motivation to create a nanoluciferase reporter was that, in our experience, luminescence is a reliable way to quantify promoter activity from in vivo samples such as feces, thanks to its low background noise. We designed a nanoluciferase reporter because we were interested in using the same reporter for in vitro and in vivo assays (shown in Fig. S3b).

In contrast, we chose a gfp reporter for the quantification of P_{glpT} activity, despite the greater amount of background signal for in vitro samples, because we were interested in using flow cytometry to test if P_{glpT} expression is bimodal in vivo (which was not the case, see Fig. S4). We took measures to minimize the background gfp signal in our flow cytometry gating strategy (see Material & Methods).

We have added a comment in the manuscript describing our rationale for choosing nanoluc as the reporter gene for P_{pstS} activation:

190 We did not knock out *pstS*, as it influences P_{pstS} activity, **and chose the nanoluc reporter gene, since it is highly compatible for use in vivo, allowing us to use the same reporter in subsequent animal experiments (see Fig. S3b)**^{11,13–15}.

b. Genes with apparently redundant functions can play roles in specific ecological contexts (for instance PMID: 29259085, PMID: 30963998 or PMID: 31753999). Stm has two additional Pi transporters (PitA and Pst), which, as far as we know, are more efficient in importing Pi than GlpT (PMID: 3522583, PMID: 6998957; PMID: 11489853). While is it feasible that GlpT is more efficient in certain environments, like the gut, the authors have not made this point.

Response. Thank you very much for this helpful suggestion. In addition to referring to these transporters as *S. Tm*'s "canonical P_i transporters" and explaining why GlpT could have an effect in the gut, we now added a sentence to strengthen this point:

Line 334

"PitA and PstS are believed to be more efficient at importing P_i than GlpT, during in vitro growth^{16–18}**".**

c. GlpT has been shown to import Pi (PMID: 3522583) and balance intracellular Pi levels in the bacterial cytosol (i.e. it exports intracellular Pi when glycerol-3-phosphate is over-imported via the Ugp system (e.g. see ref 26)). Rather than repeating the former point, the authors need to emphasize that this protein can promote the accumulation of Pi into the bacterial cytoplasm through non-equimolar Pi exchanged (PMID: 15556940 and ref 28).

Response: We agree. As far as we know, this had not previously been shown for *S. Tm*. In addition, GlpT is often described as a Gly3P importer and P_i exporter. Thus, we felt that a demonstration of the direction being reversible would be an important argument for the presented manuscript.

We highly appreciate your comment on the need to better emphasize GlpT's capacity to accumulate P_i by non-equimolar P_i exchange! Please find the changes we made in the figure legends for Fig. 3a and in the text below.

Figure legend:

Figure 3 GlpT can act as a phosphate importer

a The three known transport modes of GlpT, depending on the availability of Glycerol-3-phosphate (G3P) and P_i . GlpT's G3P and P_i antiport is bi-directional, in a concentration-dependent manner, and GlpT can additionally import P_i by non-equimolar exchange of $2P_i:1P_i$ in *E. coli*. The different numbers of representative P_i and G3P molecules aim at representing different

periplasmic/cytoplasmic concentrations. Created with BioRender.com.

Results section:

183 While traditionally known for importing G3P and exporting P_i , the GlpT transporter is bidirectional and can additionally import P_i by the non-equimolar exchange of P_i in *E. coli*^{19,20}.

Discussion section:

340 While GlpT has mostly been described for the import of G3P in exchange for P_i , the reverse transport direction was discussed several decades ago¹⁹. Additionally, GlpT has been shown to be able to import P_i by non-equimolar exchange in *E. coli* (visualized in Fig. 3a)^{19,20}.

d. Reference 41 reports that *pstS* transcription is active intracellularly, when *Stm* is replicating inside a macrophage, not that *pstS* is a reporter for intracellular P_i in the bacterium. The inference that *pstS* is transcribed in response to a decrease in intracellular P_i has been proposed by other studies (PMID: 36993483; PMID: 2943772).

Response: Thank you for pointing that out! We agree and have added references to the recent elegant studies showing that P_{pstS} responds to intracellular P_i levels in *S. Tm*^{13,15}. Unfortunately, we couldn't find the PMID: 2943772 reference and would appreciate a specification of the study being referred to.

6. Lines 160-178, the inference from indirect data using *Stm* mutants of known pathways does not convince this reviewer that glycerol/glycerol-3- P_i metabolism is not active during luminal growth. Glycerol-3- P_i can be generated by *GpsA*, which is not considered in these experiments.

Additionally, other unknown enzymes could promote glycerol/glycerol-3-Pi metabolism in this context.

Response: Thank you for your helpful comment! We fully agree with you that Gly3P metabolism can and likely must still be occurring, given its importance for phospholipid biosynthesis. The message that we wanted to convey was that the fitness effect seemed to be specific to GlpT or rather was not explained by other mutations in the G3P pathway.

We modified the text with the aim of conveying that message:

Line207 “To investigate if ~~the glycerol is metabolized during luminal growth~~ the luminal fitness advantage of the *glpT* knockout strain is related to G3P metabolism, we constructed a library of isogenic knockout strains lacking genes encoding enzymes that have been implicated in glycerol metabolism (visualized in Fig. 4a)^{21,22}.”

If bacteria were to accumulate glycerol-3-Pi during luminal growth, the presence of GlpT could reduce fitness by promoting the exchange of this metabolite with Pi present in the gut lumen.

Response: As you point out, it is possible that the negative effect of *glpT*-expression is tied to a decrease in intracellular G3P, due to G3P export. We considered that the export of Gly3P for P_i import could contribute to the fitness effect of GlpT. Since we did not have a way to experimentally differentiate the fitness effect of P_i import and Gly3P export, we placed this consideration in the discussion.

Please refer to lines 282-285 in the original manuscript or lines 384-387 in the revised manuscript:

Line 384 “The luminal disadvantage for *glpT* expression could have multiple non-exclusive reasons. If P_i import by GlpT is coupled to G3P export, the reduction in the important metabolite could reduce fitness. In addition, the P_i import in a niche with such a high phosphate availability could also pose a risk for phosphate intoxication.”

What happens if mice are fed a diet rich in glycerol-3-phosphate? Does the *glpT* locus become selectively neutral?? Does the wild-type strain become fitter than the *glpT* mutant?

Response: The proposed experiment is very interesting, and we have conducted a similar experiment by supplementing the drinking water of mice with 1 % G3P.

As you hypothesized, we found that the normalized ratio between the *glpT* mutant and the WT significantly decreased when we supplemented with G3P. The effect did not go so far as to give the WT a competitive advantage over the mutant (which would result in a normalized ratio with a median smaller than 1). We think that the reason could be the high amount of P_i in the gut. Additionally, some of the G3P might have degraded. We have added the data to the manuscript in **Fig. 2b**.

Figure 2b The fitness effect of *glpT* is linked to the high phosphate availability in the murine cecal lumen

a The fitness advantage conferred by the *glpT* mutant is reduced when mice are fed chow with an eightfold lower phosphate concentration (0.1%). The different shapes for the data points indicate that they originate from four independent experiments. The statistical significance was tested with a two-sided unpaired t-test. Lines represent the median. $n = 12$ mice, combined from four independent experiments. Please note that four data points for the regular 0.8% P_i chow control are also shown in panel 2b, as the same mice served as the control for both experiments for the sake of 3R; these data points are shown as open circles. **b** The fitness advantage conferred by the *glpT* mutant is reduced when mice are fed 1% G3P via their drinking water. The statistical significance was tested with a two-sided unpaired t-test. Lines represent the median. $n = 7$ mice. The different shapes indicate they originate from two independent experiments. Please note that four data points for the no supplementation control are also shown in panel 2a, as the same mice served as the control for both experiments to reduce the number of animals. These data points are shown as open circles. **c** [...]

7. The sentence in lines 277-231 is unclear. If the host restricts P_i availability, a mutation in other transporters should impact intracellular growth. Accordingly, a deletion of the *Pst* transporter has been shown to lower *Stm* fitness in macrophages (Ref. 41). Based on this, we can infer that the fitness of the $\Delta 3P_i$ strain would be even lower.

Response: Thank you for raising that point. Referring to lines 227-231 we were first surprised by the higher replication rate of the $\Delta 3P_i$ strain, based on the logic you outline.

The phenotype of the $\Delta 3P_i$ strain having a higher replication rate was reproducible, and is in line with the

data published by Choi et al, Nat. Comms., 2019 (<https://doi.org/10.1038/s41467-019-11318-2>) and Yang et al, Int. Journal Mol. Sci., 2023 (<https://doi.org/10.3390/ijms242417216>). The first study shows an increased replication rate for a *phoB* mutant in J774 A.1 cells²³. The *phoB* mutant should be deficient for the expression of *phoB*-regulated genes, including the PstSCAB transporter^{13,15}. The second study shows that *S. Tm*'s fold replication in HeLa cells was lower when the eukaryotic PIT1 was targeted by siRNA and higher when the eukaryotic PIT1 was overexpressed²⁴. Both studies indicate that a lower P_i availability increases *S. Tm*'s replication rate inside macrophage cells. It thus seems like the effect of P_i availability to *S. Tm* on intracellular is currently unclear. We speculate that a lower P_i availability could positively affect intracellular replication for example due to the P_i-dependent regulation of SPI-2 gene expression. Earlier SPI-2 expression within macrophages could thereby promote intramacrophage replication. However, this speculation is not based on data and should be experimentally investigated. Since it does not relate to the main messages of the presented manuscript, we propose that the dependency of intracellular replication on P_i availability be investigated in future studies.

We highlighted this open question in the discussion. This is the paragraph we added to the text:

Line 393 “An interesting side observation was the heightened replication rate of the $\Delta 3P_i$ strain in Raw264.7 macrophages (**Fig. 6d**). The finding was in line with a prior publication, which found that a *phoB* deficient *S. Tm* strain showed a higher replication rate in J774 macrophages²³. A similar conclusion can be drawn for replication in HeLa cells, where phosphate availability was inversely correlated with *S. Tm*'s replication rate²⁴. The authors knocked out the eukaryotic PIT1 gene, which should increase phosphate availability to *S. Tm* and saw a reduced replication rate for *S. Tm*²⁴. Accordingly, overexpression of PIT1 increased *S. Tm*'s replication rate²⁴. In contrast, a different study found that a *phoB*- and *pstSCAB*-deficient *S. Tm* strain was attenuated in replication in HeLa cells and Raw264.7 macrophage cells²⁵. The conflicting results show that the effect of phosphate availability and lack thereof at intracellular sites on *S. Tm* fitness is unresolved and should be addressed in future studies.”

8. Lines 239-241, the increase in *gloC* promoter activity is consistent with the proposed role of *GlpT* in methylglyoxal detoxification. However, what happens to the fitness/replication of the *glpT* vs wild-type strain in a background that is impaired for methylglyoxal detoxification (e.g. *mgsA*, *gloB* or *gloC*)? Does the presence of *GlpT* become irrelevant in a *mgsA*, *gloB* or *gloC* background (ref. 51)?

Response: Thank you for inquiring. That is a great suggestion, especially to differentiate between an effect between the importance of Gly3P metabolism and GlpT's proposed function in methylglyoxal detoxication/survival of oxidative stress. We conducted experiments with strains deficient for *mgsA*, *gloA*, and *gloB* as single knock-outs, and in combination with the *glpT* knockout. We did not include *gloC* deficient strains since there is a second putative *gloC* homolog encoded in the SL1344 genome, and it is unclear if it is redundant in function. We found that all strains were attenuated for replication in Raw264.7 cells (replication rate of ~1 compared to ~10 for the WT control). As a result, the replication assay did not have sufficient resolution to disentangle a methylglyoxal-dependent and -independent effect of *glpT*, even when we combined data from three separate experiments (n = 9) to increase the statistical power. We added the new data to the manuscript as supplementary Figure 5 and described it in the text (see Fig. S5).

Figure S5 It remains unclear if GlpT contributes to replication in Raw264.7 cell in methylglyoxal-dependent and -independent ways

The comparison of the replication rates for the *glpT* knock-out strain to *mgsA::cat*, *gloA::cat*, and *gloB::cat* deficient mutant strains does not have sufficient statistical power to discriminate whether *glpT*'s contributions to intramacrophage replication are methylglyoxal-dependent and -independent. The statistical significance was tested by an ordinary one-way ANOVA with Tukey correction. The lines indicate the median. n = 12 Raw264.7 macrophage infections, combined from four separate experiments.

Changes in the manuscript:

Line 294 To discriminate between a methylglyoxal-dependent effect, such as a contribution to resistance against oxidative stress, and a methylglyoxal-independent effect, such as import of G3P as a carbon source, we compared the replication rate of the *glpT* deficient strain, to the replication rate of strains deficient for only *mgsA*, *gloA*, and *gloB* or in combination with the *glpT* knock out. MgsA is responsible for the formation of methylglyoxal, while GloA and GloB are involved in its detoxification (see **Fig. 4a**)^{26–28 29,30 27,30,31}. *mgsA*, *gloA*, and *gloB* deficient strains should all be less protected against oxidative stress than WT *S. Tm*³². The replication rate was similar for all strains (around 1), indicating that our replication assays do not have sufficient resolution to be able to disentangle GlpT's contribution to replication in Raw264.7 macrophages in methylglyoxal-dependent and/or -independent ways (**Fig. 6e**), even when we combined datasets from three separate experiments, to increase the statistical power (**Fig. S5**).

9.Lines 292-303:

a. If the GlpT decreases fitness in the lumen by importing excess of Pi into the cytosol of Stm, would over-expression of the Pi exporter YibB (PMID: 2148893) rescue this phenotype?

Response to **a**: Thank you for this interesting question. We believe that could well be and represent a third way one could demonstrate that the fitness effect of *glpT* depends on P_i availability in addition to the experiments with diets with different P_i contents and the Δ3P_i strain shown in Figures 2a and 2b, respectively. In a recent study, the induction of *yjbB* overexpression from a plasmid led to a slight decrease in in vitro bacterial growth measured as OD_{600nm}¹³. Even a small growth defect in vitro could lead to significant attenuation in our mouse model, which could additionally select for the loss of the plasmid or mutations. Therefore, we believe that using a YjbB overexpression plasmid in vivo could be less reliable than using the modified diet and the Δ3P_i strain background we used here.

b. The really puzzlingly fact about the data is that given that *glpT* can be regulated (https://bioinf.gen.tcd.ie/cgi-bin/salcom.pl?header_rotation=45;query=glpT;db=salcom_mac_HL), why is this protein being expressed in the lumen (or is it)? Does *Stm* inhibit *GlpT* expression, but the small amount of *GlpT* that is being produce is detrimental? This phenomenon is reminiscent of other studies that have shown that *Escherichia coli* accumulates mutations in regulated proteins as it passes through the mammalian gut (PMID: 21698140).

Response to **b**: Great question! It really is puzzling, and a topic we're currently very interested in.

glpT is very likely expressed in the intestinal lumen since we can detect much higher fluorescence when *S. Tm* carries a P_{glpT}-controlled *gfp* fluorescence reporter plasmid compared to the control *S. Tm* strain that carries the promoter-less plasmid (see days 1 and 4 post-infection, **Fig. S4c**).

This still leaves us with the question why *S. Tm* expresses a gene in the gut lumen, that can reduce its fitness in the environment of the mouse gut lumen. We believe that in the presence of strong competition in the intestinal lumen (such as in the presence of a complex microbiota), will normally limit *S. Tm*'s growth in the gut lumen of most hosts. In these common cases, *S. Tm* benefits from quickly accessing extra-luminal niches such as the lamina propria, where the pathogen would rely on intracellular replication. This could explain how genes enhancing the fitness of intracellular growth are maintained, even if their gut-

luminal expression might be of disadvantage under some circumstances. Other circumstances, where *S. Tm* chronically persists at systemic sites, could also select for genes important for intracellular replication. In contrast, infections of an individual with a disrupted microbiota, where the luminal *S. Tm* population becomes very large, could select for inactivating mutations in *glpT*. This would be attributable to the large population size per se (which increases the likelihood of *glpT* disrupting mutations to occur) and the gut-luminal pathogen growth, which would select of such mutants over the ancestral wildtype. Such pronounced gut-luminal growth is a characteristic of our streptomycin colitis model³³. These considerations are why we consider your previous question regarding if *glpT* mutations are found in *S. Tm* colonizing mice harboring a microbiota highly relevant. The existing evidence suggests that transient alleviation of colonization resistance by food, antibiotics or other stressors may alter the gut-luminal niche to select for mutations in particular genes. Such niches may also explain the data by Cherry² which indicates that conditions in which loss of function of *glpT* is selected do arise often enough to be picked up in when analyzing 100'000 genomes. In either case, in order to allow for positive selection of *glpT*-deficient mutants, the *glpT* gene must be expressed by the parental strains in the respective niches.

Context-dependent selection for mutants has also been observed for other enterobacteriaceal genes. The expression of the virulence regulator *hilD* of *S. Tm* is one prominent example. Here, costs of expression in the gut lumen result in the selection of *hilD*-inactivating mutations when the pathogen grows over extended periods of time at high densities within the lumen of a microbiota-depleted inflamed gut^{3,34}. Similarly, gut luminal growth of *E. coli* K12 in germ-free mice selected for mutants in flagella, *ompB* and the maltose regulator gene *maltT*³⁵. Another study found that gut-luminal growth can select for the co-existence of *E. coli* strains which can utilize galactitol, and *gat* mutants which cannot³⁶. Bioinformatics analysis of carbohydrate utilization genes in Enterobacteriaceae provided evidence that such genes might be under significant context-dependent selection pressures, resulting in substantial differences in carbohydrate utilization gene expression between different Enterobacteriaceae³⁷.

Based on these observations, we have started a project, where we compared the *glpT* expression between different *S. Tm* strains. We found that another commonly studied *S. Tm* strain ATCC14028, does not express *glpT* in the intestinal lumen (quantified using the same P_{glpT}-GFP reporter by flow cytometry on fecal samples, see panel **b** in the graph below). As a result, *glpT*-deficient ATCC14028 does not have a competitive advantage against its isogenic WT in the intestinal lumen (see panel **a** in the graph below).

Please note that the data displayed below is part of a follow-up study that has not yet been completed, which is why it is not included in the present manuscript. We have instead modified the text of the discussion section to alert the readers to other examples of the selection for gene-inactivating mutations and the evidence for differences in carbohydrate utilization gene regulation across the Enterobacteriaceae. We expanded the discussion in the manuscript by making the changes highlighted in red:

[REDACTED]

Line 420 Alternatively, the selection of *glpT* mutants observed in our study may result from reduced levels of colonization resistance which are characteristic for the mouse models that we have used ³³. This should enhance the chances of selecting for mutants with enhanced fitness in such low-colonization-resistance niches. Transient alleviation of colonization resistance by food, antibiotics or other stressors may alter the gut-luminal niche to select for mutations in particular genes also in natural *Salmonella* populations, explaining why loss of function of *glpT* and other several dozen other genes are selected to arise often enough to be picked up in when analyzing 100'000 genomes ². Positive selection of *glpT*-deficient mutants, is only possible when the *glpT* gene is expressed by the parental strains in the respective niches.

Context-dependent selection for mutants has also been observed for other enterobacteriaceal genes. The expression of the virulence regulator *hilD* of *S. Tm* is one prominent example ^{3,34}. Here, costs of expression in the gut lumen result in the selection of *hilD*-inactivating mutations when the pathogen grows over extended periods of time at high densities within the lumen of a microbiota-depleted inflamed gut. Similarly, gut luminal growth of *E. coli* K12 in germ-free mice selected for mutants in

flagella, *ompB* and the maltose regulator gene *malT*. Another study found that gut-luminal growth can select for the co-existence of *E. coli* strains which can utilize galactitol, and *gat* mutants which cannot³⁶. Bioinformatics analysis of carbohydrate utilization genes in *Enterobacteriaceae* provided evidence that such genes might be under diverse context-dependent selection pressures, resulting in substantial differences in the control of carbohydrate utilization gene expression between different *Enterobacteriaceae*³⁷. In conclusion, our data indicates that *glpT* mutations are selected in the luminal niche, resulting in a disadvantage in the phagocytic and potentially other phosphate- and nutrient-limited niches. The counterselection follows much slower kinetics, as the ratio reaches 10 within the first day of infection but takes the following three days to decline to the same extent. Yet, the data suggest that the capacity for intramacrophage replication has a crucial effect on the luminal population and, by extension, fecal shedding and infection of new hosts.

Reviewer 2

Based on the reasonable working hypothesis a part of which is supported by their previous report, the author assessed the fitness effects of glpT mutations in S. enterica during multiple phases of orogastric mouse infection. The authors found that glpT-deficient mutants outcompete WT at initial growth in the gut lumen when phosphate availability is high. They also showed that the ratio of glpT-deficient mutants and WT changed at the later phase depending on macropases. Their findings may be interesting, but some conclusions are not clearly evident and/or explained. Also, the writing in the manuscript needs to be improved.

Response:

Dear Reviewer 2,

Thank you for your positive assessment and for your constructive criticism! We particularly appreciate your help making the text more accessible to readers.

Thank you and all the best,

Wolf, Noemi and the rest of the team

Major comments:

(1)

One of the keys of this work is that the author found a metabolic trade off for the fitness. Because other trade-off genes such as T3SS and flagellin are known, the author should describe more how the discovery of a metabolic gene is significant, probably in the introduction.

Thank you for pointing out that we could further stress the significance of our findings.

As you point out, the trade-off for the known examples such as the T3SS and flagellins can be attributed to the host's immune system. Thus, the nature of the antagonistic pleiotropy is somewhat straightforward. Nevertheless, these examples have greatly shaped our understanding of the evolution of pathogens and contributed to the ideation of therapeutic approaches. For example, understanding that antagonistic pleiotropy affects the evolution of *S. Tm*'s genes encoding surface-exposed structures helped explain why conventional vaccination attempts have been mostly unsuccessful (described in multiple recent reviews such as ³⁸⁻⁴¹). On a positive note, the understanding allowed us to create a new form of vaccination termed evolutionary trap vaccine or EvoVax, which works by exploiting that escape variants are selected for by the immunized host immune system are attenuated ^{42,43}. The proof of concept for the approach has been successfully shown in mice and pigs and has gotten positive feedback from stakeholders such as veterinary pharmaceutical companies interested in developing it as a tool to reduce the pathogen burden in (stock) animals, which could have major implications for public health ^{42,43}.

In contrast, the concept of antagonistic pleiotropy acting on metabolic genes of bacterial enteric pathogens due to the strong differences in host niches has, as far as we know, not been widely appreciated. One previously identified example of a group of metabolic genes with a highly context-dependent fitness effect are the genes encoding enzymes involved in galactitol utilization in *E. coli* ^{36,44}. Interestingly the authors showed that polymorphism, i.e. functional and non-functional variants of the genes, seems to be selected ³⁶. A significant amount of research on the galactitol utilization has revealed that the apparently opposing selective pressures for or against the *gat* operon are likely the result of the sugar-alcohol being rare and available in different concentrations in different diets. As a result, *Enterobacteriaceae* capable of metabolizing galactitol have a competitive edge when galactitol is present and if other more favorable carbon sources are subject to strong competition by other microbes, making it a great example for the restaurant hypothesis ^{36,44-49}. The currently presented example is different as it highlights that *glpT* seems to be subject to selective pressures for niche specialization (i.e. the gut lumen vs the host's tissue), which has previously been proposed as a potential result of antagonistic pleiotropy in the field ^{50,51}. In a study recently published in *Nature* the authors found evidence for such niche specialization resulting from conflicting selective pressures in the pathobiont *E. gallinarum* ⁵². *E. gallinarum* evolved two separate lineages that specialized in colonizing the intestinal lumen or translocating to the liver, respectively, indicating a luminal vs. a mucosal specialization ⁵². Similarly, our data suggests that *glpT*-deficient cells have an advantage during luminal growth, and *glpT*-proficient cells have an advantage inside mucosal macrophages. Mucosal-adapted *E. gallinarum* were better at persisting in the liver in vivo, just like *glpT*-proficient *S. Tm* seem to be better equipped at infecting the spleen ^{9,52}. While the luminal adaptations in mannose and lactose utilization identified in *E. gallinarum*, were previously known to be important for gut luminal colonization, P_i antiport being of a disadvantage in the face of the high P_i availability in the gut has, to our knowledge, not been described before and provides a new concept that while P_i is limiting at many systemic sites, its high availability in the intestinal lumen, poses a risk. The study by Cherry suggests this affects many different *Salmonella* strains, and our conservation analyses indicate that GlpT presence and function is highly diverse among pathogenic members of the *Enterobacteriaceae*, suggesting that our study could be relevant well beyond *Salmonella* spp., which infect millions of people and many other animals every year across the globe.

We have expanded our introduction in the manuscript to allude to the most important points mentioned here.

Changes in the manuscript:

Line 83 Understanding how a gene can increase or decrease a pathogen's fitness in a context-dependent fashion is important for developing effective strategies to control infectious diseases and understanding the evolution of pathogens with and without interventions such as antibiotic use or vaccination. For example, understanding that antagonistic pleiotropy affects the evolution of *S. Tm*'s genes encoding surface-exposed structures helped explain why conventional vaccination attempts have been mostly unsuccessful (recently reviewed in ³⁸⁻⁴¹). On a positive note, the understanding allowed us to create a new form of vaccination termed evolutionary trap vaccine or EvoVax, which works by exploiting that escape variants are selected for by the immunized host immune system are attenuated ^{42,43}.

So far, most examples of antagonistic pleiotropy have been attributed to the risk of detection by the host's immune system. Given that nutrient, mineral, and electron donor or acceptor availability differ considerably between the luminal and intracellular lifestyles, we hypothesized that some metabolic genes should similarly have fitness effects that are highly lifestyle-dependent and face opposing selective pressures. So far, there is very little information about the role of phosphate availability during gut infection. Evidence for antagonistic

pleiotropy in metabolic genes adds to our understanding of the evolution of bacterial pathogens, with likely implications for preventative and therapeutic interventions.

(2)

The author showed that the fitness of *glpT*-deficient mutants in *S. enterica* changed over time (day 0-day 4), and also mentioned that this change is due to the change of environment from “in the gut lumen” to “inside phagocytes”. The reviewer thinks that the author showed that this change is dependent on the macrophage, but did not demonstrate that *S. enterica* cells are actually present in the macrophages.

Response: That is a very good point. Intramacrophage survival and replication are considered a key feature of *S. Tm*'s pathogenesis^{53–55}. Our lab has previously shown that in the lamina propria, *S. Tm* is only transiently located extracellularly before co-localizing with macrophages in microscopic images⁵⁶. Flow cytometry data, such as data we recently published, confirmed that the *S. Tm* cells are inside the macrophages⁵⁷. Therefore, there is sufficient evidence from prior studies that *S. Tm* is located within macrophages. We have additional unpublished flow cytometry data recapitulating what we found and published in PNAS last year (see⁵⁷), which we could share confidentially.

To clarify that, we have added a sentence in the text, highlighting intra-macrophage growth as a hallmark of *S. Tm*.

Line 271 In the lamina propria, many *S. Tm* cells are lodged within macrophages and *S. Tm* can replicate in tissue-resident macrophages, a P_i -scarce environment^{25,58}

(3) L116-L129

It is not easy to understand how the results shown in Fig.2a, b, d can be explained in a consistent manner. The reviewer suggests that the authors clearly explain possible interpretation(s) for these results step by step. This may help readers for subsequent reading.

Response: Thanks for pointing that out! We have modified the text accordingly and hope the new version is easier to follow. Our two takeaways for the figure are:

- The normalized ratio between the *glpT*KO and its parental strain depends on phosphate availability for an unknown reason (**Fig. 2a-b**). More specifically, the *glpT* knock-out strain outcompetes the wildtype on day 1 p.i. when phosphate availability is high, but not when it is lowered.
- The phosphate availability to *S. Tm* in the intestinal lumen is so high that the pathogen is not attenuated when its canonical P_i transporters are knocked out (**Fig. 2c-d**).

These are the main changes we made in the manuscript, highlighted in red:

Line 140 “*GlpT* is a G3P and P_i antiporter with transport activities that depend on G3P and P_i concentrations¹⁹. Because of that, we wondered if the competition between the *glpT* mutant and the WT depends on substrate availability^{15,19}. To test that, we wanted to modulate the substrate availability to *S. Tm* in the intestinal lumen in three independent ways. First, we procured mouse chow with a reduced P_i content of 0.1 % as one way to reduce the phosphate availability to *S. Tm* in the intestinal lumen. We then compared the normalized ratio between the *glpT* mutant and the WT in mice fed the regular 0.8 % P_i

chow, to the normalized ratio between the *glpT* mutant and the WT in mice fed the 0.1 % P_i chow. The diet with a reduced P_i content significantly reduced the advantage of the *glpT* knock-out (KO) on day 1 p.i., to a normalized ratio of around 3 (Fig. 2a). This suggested that the fitness effect of *glpT* depends on the transport intensity and/or direction, which depend on substrate availability. As a second way to influence substrate availability, we supplemented 1% G3P in the drinking water of the mice (Fig. 2b). Similar to the reduction of P_i availability (black data points, Fig. 2a), this should lead to more G3P import and P_i export, which should increase the fitness of the *glpT*-proficient WT. As expected, the G3P supplementation too significantly lowered the normalized ratio between the *glpT* knockout strain and the isogenic WT (Fig. 2b). As a third way to validate that the fitness effect of *glpT* depends on substrate availability, we knocked out all three known P_i transporters encoded by *S. Tm* ($\Delta 3P_i$; *pitA*, *pstS*, *yjbB*)^{38–41}. The resulting strain ($\Delta 3P_i$; *pitA*, *pstS*, *yjbB*) likely has less access to phosphate than the WT.”

Line 160 “Next, we tested the relative fitness of the *glpT* KO in the $\Delta 3P_i$ background compared to the parental strain ($\Delta 3P_i$). Similarly to the low P_i diet, the low availability resulting from this genetic background ($\Delta 3P_i$), strongly decreased the ratio between the *glpT* KO and its isogenic parent (WT background) to around 1 ($\Delta 3P_i$ background) (Fig. 2b). The decrease in the normalized ratio between the *glpT* KO and its parental strain, when access to P_i was reduced with the modified diet, or the lack of P_i transporters, establishes a link between the fitness effect of *glpT* and phosphate availability. The normalized ratio between the $\Delta glpT pitA::cat$, $\Delta glpT pstS::aphT$, and $\Delta glpT yjbB::aphT$ double knock-out strains can be found in Fig. S2a. We additionally tested the competitive fitness of the individual KOs (*pitA*, *pstS*, and *yjbB*), visualized in Fig. 2c) and the triple KO ($\Delta 3P_i$) in competition against the WT on day 1 p.i. All of them led to a nonsignificant ratio, close to 1, even when all three transporters were knocked out in the same strain (Fig. 2d). The neutral competition suggested that the luminal phosphate availability for *S. Tm* is surprisingly high.

(4)

It seems that the manuscript has not yet been finalized: Table 2 is incomplete, Fig.S2c and S4d are not sited (might be due to mistakes). Some parts are not easy to understand because important information seems to be missing: detailed comments are provided below.

Response: Thank you very much for pointing this out. We have addressed these points in the revised manuscript (Table 2, Fig. S2c, S4d).

Other comments:

(5) L107 “differential plating”

The reviewer thinks that the authors need to describe clearly and with more details how this was performed to show that the obtained numbers or the calculated ratios are reliable. It is desired to exhibit some raw data examples.

Response: Thank you for pointing this out. We agree and have modified the sentence that introduces our means of strain quantification in the following way:

“We determined the abundance of both strains in the feces of the mice by **differential** plating **them on selective media** and calculated the ratio between the strains, normalized to the ratio in the inoculum.”

We also expanded our description in the materials and methods section:

“CFU quantification of in vivo samples was performed by plating diluted samples on MacConkey agar with appropriate antibiotics and counting. **For competitive infections with two different *S. Tm* strains, the two strains were quantified by differential plating, where the two strains were quantified after plating them on solid media containing antibiotics, which only the strain of interest was resistant to. The normalized ratio is defined as the number of CFUs for the mutant divided by the number of CFUs by the isogenic parental strain, normalized to the ratio present in the inoculum.**”

All the raw data, including the colony-forming units or CFUs data, will be made freely available via the ETH research collection. The data will be associated with the doi of the publication and thus can only be published by ETH’s research collection once the doi is known.

We will use our most recent data to present an example calculation of the normalized ratio. We gathered the data for an interesting new experiment suggested by another reviewer. This other reviewer was interested in finding out what would happen to the normalized ratio between the *glpT::aphT* knock-out strain and the WT on the first day of infection, if we supplemented G3P. The hypothesis was, that G3P supplementation would lead to an advantage for the WT, such that the ratio between the *glpT::aphT* knock-out strain and the WT would be lower when G3P is supplemented than when it isn’t.

Our readout after plating the inoculum and samples on MacConkey agar containing Kanamycin for the *glpT::aphT* strain and Chloramphenicol for the Chloramphenicol-resistant wild-type (WT CmR) are the colony-forming units or CFUs we count. We then multiply the number of CFUs by the dilution factor. For in vivo samples such as feces, we weigh the empty and full tubes to get the number of CFUs per gram of feces.

This is what the CFU data for the mentioned experiment looks like:

From the CFU data we calculate the normalized ratio as follows:

Number of CFUs from fecal samples growing on MacConkey agar with Kanamycin (*glpT::aphT*, KanR) and dividing it by the number of CFUs growing on MacConkey agar with Chloramphenicol (WT, CmR). We then divided the ratio for the in vivo samples by the ratio of the two strains in the inoculum to normalize it, resulting in what we call the “normalized ratio”. The normalized ratio is often referred to as competitive index in the field. We decided to refer to it as normalized ratio as we felt that the term was more descriptive and, therefore, accessible to people unfamiliar with this readout. We have added a comment in the text that the normalized ratio is the same as the competitive index, in the text.

This is how the calculation looked like for the two first mice of the respective groups:

Ratio for *glpT::aphT* (KanR) on day 1 p.i. compared to the WT (CmR) **without** supplementation of G3P in the drinking water:

$1.65e9 \text{ KanR CFUs} / 1.70e8 \text{ CmR CFUs} = \text{ratio of } 9.69$

Ratio in the inoculum: $7.10e8 \text{ CFUs (} glpT::aphT) / 6.53e8 \text{ CFUs (WT CmR)} = 1.09$

Normalized ratio (ratio in vivo sample divided by ratio in the inoculum):

$9.69/1.09 = \mathbf{8.91}$

In contrast, this is what the ratio for *glpT::aphT* on day 1 p.i. compared to the WT (CmR) **with** supplementation of G3P in the drinking water for the first mouse looked like:

$1.38e9 \text{ KanR CFUs} / 7.22e9 \text{ CmR CFUs} = \text{ratio of } 1.92$

Ratio in the inoculum: $7.10e8 \text{ CFUs (} glpT::aphT) / 6.53e8 \text{ CFUs (WT CmR)} = 1.09$

Normalized ratio (ratio in vivo sample divided by ratio in the inoculum):

$1.92/1.09 = \mathbf{1.76}$

We introduced a more detailed description in our materials & methods section. These are the changes we made (highlighted in red):

CFU quantification of in vivo samples was performed by plating diluted samples on MacConkey agar with appropriate antibiotics and counting. **For competitive infections with two different *S. Tm* strains, the two strains were quantified by differential plating, where the two strains were quantified after plating them on solid media containing antibiotics, which only the strain of interest was resistant to. The normalized ratio is defined as the number of CFUs for the mutant divided by the number of CFUs by the isogenic parental strain, normalized to the ratio present in the inoculum.**

Please find examples in the literature of the *Salmonella* field to see how our colleagues present such data (which they refer to as competitive index). Here are three recent examples from different established groups: Yoo et al, Cell Host & Microbe, 2024 (<https://doi.org/10.1016/j.chom.2024.05.001>); Kim et al, Cell Host & Microbe, 2024 (<https://doi.org/10.1016/j.chom.2024.01.004>); Osbelt et al, Nature Microbiology, 2024 (<https://doi.org/10.1038/s41564-024-01710-0>)

(6) L109 “Notably, the WT outcompeted from day 2 and the mutant/WT ratio progressively declined

110 from around 10 to around 1 by day 4 p.i. (Fig. 1b).”

Is this tendency conserved within individual mice? Or some mice showed different patterns?

The reviewer suggests that the multiple dots at different timepoints from same mouse are linked.

That is a good point; thank you for raising it. We have included connecting lines between the samples gathered at different time points from the same mice and updated the figure (see below). The graphs shows that the trend is consistent for most of the mice, with a few mice showing a different pattern, such as a higher normalized ratio of the *glpT* knockout strain vs. its isogenic WT by day 3 p.i., before the normalized ratio declines by day 4 p.i.

(7) L122

Current Fig. 2c should be cited here and labeled as Fig. 2b.

Response: Thank you.

(8) L143

It is not clear how the three scenarios are selected to show.

(why other scenarios are not illustrated?)

Also, the meaning of illustrations are not clear:

e.g., why does not scenario 3 show Gly-3-P?

why is the number of Gly-3-P and Pi different between scenario 1 and 2?

Response: Thanks for asking. These are the three modes of transport that have been described in the literature for GlpT.

G3P is not shown in the third scenario because in this transport mode, GlpT exchanges 2 Pi molecules for 1

P_i

molecule.

The different numbers of G3P / Pi molecules in the first and second scenarios symbolize different periplasmic/cytoplasmic concentrations.

We have expanded the figure legend and added a sentence on the mechanism shown in “Scenario 3” to the manuscript.

Line 183 “While traditionally known for importing G3P and exporting P_i , the GlpT transporter is bidirectional and can additionally import P_i by the non-equimolar exchange of P_i in *E. coli*^{31,46}. Thus, P_i import should be the main transport direction when the extracellular P_i concentrations are considerably higher than G3P (visualized in Fig. 3a, scenarios 2 and 3)^{26,31}.”

Figure Legend:

The three known transport modes of GlpT, depending on the availability of Glycerol-3-phosphate (G3P) and P_i . GlpT's G3P and P_i antiport is bi-directional, in a concentration-dependent manner. In the first hypothetical scenario, there is more G3P than P_i in the periplasm and the main transport direction is G3P import and P_i export, by equimolar G3P: P_i antiport. In the second hypothetical scenario, there is more P_i than G3P in the periplasm, and the main transport direction is P_i import and G3P export, by equimolar P_i :G3P antiport. GlpT can additionally import P_i by non-equimolar exchange of 2 P_i :1 P_i in *E. coli*, which is

shown in the third scenario. The different numbers of representative P_i and G3P molecules aim at representing different periplasmic/cytoplasmic concentrations. Created with BioRender.com.

(9)

L147

Sometimes “mutant” is not clear.

Can the authors use, e.g., “knock out” or something related?

Response: We appreciate this concern, as the term “mutant” can refer to isogenic knockout strains as well as to natural isolates naturally lacking a particular gene or carrying a particular inactivating mutation. We have tried our best to make it clear what is meant in each case.

(10)

L153 “final”

What does the “final” mean?

If this is a time course measurement, the authors need to define the time “0 h”.

Response: We replaced the word “final” with “at 24 h”, to make it more specific. We refer to time 0 h as the time when the overnight culture was diluted in the medium. We have expanded the description of the assay in the materials and methods section:

Line 522 “For the overnight growth, 1 mM P_i in the form of K_2HPO_4 was added to the medium. In the morning, cells were diluted 1:100 in the same medium. After 4 h, cells were washed and resuspended in the base medium free of added P_i with a titration of K_2HPO_4 at the indicated concentration. The time of resuspension in the medium with the addition of the indicated P_i concentration is considered timepoint 0 h. The cultures were incubated at 37 °C in aerobic conditions and with shaking. After 24 h, the OD_{600nm} was measured, the cells were pelleted and frozen, until the luciferase assay was conducted. The reporter activity was normalized to the OD_{600nm} . “

(11)

L155

The reviewer suggests that the author specifically mention what the “environmental context” are. Also, how is this statement related to the main conclusion in this study?

Response: Thank you for raising that. Based on the literature, relevant environmental parameters on GlpT’s transport activity likely include G3P and P_i concentrations, pH, magnesium availability, the presence of reactive oxygen species, the expression transport activity of PstCAB, and the transport activity and direction of PitA and YjbB. For the referenced sentence, the most relevant “environmental context” is likely G3P and P_i availability.

We changed the sentence accordingly (changes highlighted in red):

Line 199 Thus, GlpT’s transport activity depends on environmental ~~context~~ parameters such as the G3P and P_i availability, and GlpT might contribute to P_i import during luminal growth, a milieu where we found phosphate availability to be high (**Fig. 2d** and **S2a**).

We want to stress that we cannot, at this stage, determine the relevance and contributions of the named factors nor rule out that other factors influence GlpT’s phenotype. This study’s main conclusion is that GlpT’s transport activity and effect on *S. Tm*’s fitness depends on the environmental context, with a focus on the diet (high P_i vs low P_i) and lifestyle (luminal vs intracellular, where the latter is thought to have low P_i availability). The context-dependency was our main motivation for this study and its title, which was why we highlighted it in the referenced sentence.

(12)

L156 “The reduction in the fitness advantage...”

Is this the authors experimental results?

If so, please site a figure or something which supports this statement.

Response: Thank you for pointing out that the references to Figures 2a and 2b were missing. We have modified the text in the manuscript accordingly.

(13)

**L165 “Since the *ugp* operon is induced during phosphate limitation,”
Is this reported result? If so, please site a reference.**

Response: Yes, this is a result that was recently reported for *S. Tm*. The study that found the induction of the *ugp* operon during phosphate limitation in *S. Tm*, was already cited at the end of the sentence (Ref 29; Bruna et al, 2023). We moved the references to the middle rather than the end of the sentence to make it easier to spot them and added a second relevant reference for a study that was published after our initial submission ¹³.

(14)

L166 “the ratio”

What is “the ratio” specifically?

Response: Thanks for asking for clarification. “The ratio” was meant to refer to the normalized ratio between the *ugpB* knock-out strain and the WT. The sentence now reads:

Line 211 “Since the *ugp* operon is induced during phosphate limitation ^{37,43}, we **expected to find a neutral normalized ratio between the *ugpB* knock-out and the WT strain (≈ 1 , Fig. 4c) [...]**”.

(15)

L167 “The ratio for the *glpFK* mutant suggested that glycerol uptake and phosphorylation do not influence the fitness in the lumen”

This sentence is too sudden. For example, the author should explain what *glpFK* is in the text.

Response: Thank you for pointing that out. The revised version better highlights the functions of GlpF and GlpK. We intentionally kept this section short and started by mentioning the neutral effect on the ratio between the knock-out strain and the WT as we did not want readers to be feel a bit frustrated if they had read a long section and spent time and energy to understand the function of different enzymes, just to find out that they do not explain the phenotype.

In response to the comment, we have revised section in the manuscript:

Line 213 “GlpF and GlpK enable the glycerol uptake and phosphorylation respectively ⁶². The neutral ratio for the *glpFK* knock-out strain suggested that uptake via the GlpFK route does not influence the fitness in the lumen, potentially because GlpT could export the G3P ⁶³.”

(16)

e.g., Line 266-272

The reviewer suggests that some results or direct interpretations mentioned here can be described in the result section right after the corresponding experimental results, so that readers can understand and interpreted the results easier.

Response: Thank you. We are glad that the interpretations in the discussion section were helpful. We think that your suggestion is great and fully agree that readers benefit from interpretations being present in the results section. Whenever that was not the case already, we have added a concluding sentence at the end of every paragraph in the results section.

(17)

L291

In terms of equilibrium, ATP concentration may be increased because of more Pi. Please check it.

Response: Thank you for raising that point. We have checked it and confirm that as stated in the manuscript thermodynamics predict that an increase in intracellular P_i concentration reduces the ATP concentration in the cell (Ref 77: ⁶⁴). It is worth noting that P_i starvation can also lead to a reduction in ATP levels since P_i starvation is thought to lead to an accumulation of (p)ppGpp, which in turn can inhibit ATP biosynthesis ^{13,65}.

References

1. Horstmann, J. A. *et al.* Methylation of Salmonella Typhimurium flagella promotes bacterial adhesion and host cell invasion. *Nat Commun* **11**, 2013 (2020).
2. Cherry, J. L. Selection-Driven Gene Inactivation in Salmonella. *Genome Biol Evol* **12**, (2020).
3. Gül, E. *et al.* The microbiota conditions a gut milieu that selects for wild-type Salmonella Typhimurium virulence. *PLoS Biol* **21**, e3002253 (2023).
4. Grote, A. *et al.* Persistent Salmonella infections in humans are associated with mutations in the BarA/SirA regulatory pathway. *Cell Host Microbe* **32**, 79-92.e7 (2024).
5. Stecher, B. *et al.* Salmonella enterica Serovar Typhimurium Exploits Inflammation to Compete with the Intestinal Microbiota. **5**, e244 (2007).
6. Stecher, B. *et al.* Comparison of Salmonella enterica serovar typhimurium colitis in germfree mice and mice pretreated with streptomycin. *Infect Immun* **73**, 3228–3241 (2005).
7. Kaiser, P., Diard, M., Stecher, B. & Hardt, W.-D. The streptomycin mouse model for Salmonella diarrhea: functional analysis of the microbiota, the pathogen's virulence factors, and the host's mucosal immune response. *Immunol Rev* **245**, 56–83 (2012).
8. Nguyen, B. D. *et al.* Import of Aspartate and Malate by DcuABC Drives H₂/Fumarate Respiration to Promote Initial Salmonella Gut-Lumen Colonization in Mice. *Cell Host Microbe* **27**, 922-936.e6 (2020).
9. Steeb, B. *et al.* Parallel Exploitation of Diverse Host Nutrients Enhances Salmonella Virulence. *PLoS Pathog* **9**, e1003301 (2013).
10. Burton, N. A. *et al.* Disparate impact of oxidative host defenses determines the fate of salmonella during systemic infection in mice. *Cell Host Microbe* **15**, 72–83 (2014).
11. Bruna, R. E., Kendra, C. G., Groisman, E. A. & Pontes, M. H. Limitation of phosphate assimilation maintains cytoplasmic magnesium homeostasis. *Proceedings of the National Academy of Sciences* **118**, e2021370118 (2021).
12. Motomura, K. *et al.* Overproduction of YjbB reduces the level of polyphosphate in Escherichia coli: a hypothetical role of YjbB in phosphate export and polyphosphate accumulation. *FEMS Microbiol Lett* **320**, 25–32 (2011).
13. Bruna, R. E., Kendra, C. G. & Pontes, M. H. An intracellular phosphorus-starvation signal activates the PhoB/PhoR two-component system in Salmonella enterica. *mBio* **0**, e01642-24 (2024).

14. Gardner, S. G., Johns, K. D., Tanner, R. & McCleary, W. R. The PhoU protein from *Escherichia coli* interacts with PhoR, PstB, and metals to form a phosphate-signaling complex at the membrane. *J Bacteriol* **196**, 1741–1752 (2014).
15. Bruna, R. E., Kendra, C. G. & Pontes, M. H. Phosphorus starvation response and PhoB-independent utilization of organic phosphate sources by *Salmonella enterica*. *Microbiol Spectr* **0**, e02260-23 (2023).
16. Ambudkar, S. V, Larson, T. J. & Maloney, P. C. Reconstitution of sugar phosphate transport systems of *Escherichia coli*. *Journal of Biological Chemistry* **261**, 9083–9086 (1986).
17. Willsky, G. R. & Malamy, M. H. Characterization of two genetically separable inorganic phosphate transport systems in *Escherichia coli*. *J Bacteriol* **144**, 356–365 (1980).
18. Harris, R. M., Webb, D. C., Howitt, S. M. & Cox, G. B. Characterization of PitA and PitB from *Escherichia coli*. *J Bacteriol* **183**, 5008–5014 (2001).
19. Maloney, P. C., Ambudkar, S. V, Anatharam, V., Sonna, L. A. & Varadhachary, A. Anion-exchange mechanisms in bacteria. *Microbiol Rev* **54**, 1–17 (1990).
20. Hall, J. A. & Maloney, P. C. Altered Oxyanion Selectivity in Mutants of UhpT, the Pi -linked Sugar Phosphate Carrier of *Escherichia coli**. *Journal of Biological Chemistry* **280**, 3376–3381 (2005).
21. Lin, E. C. Glycerol dissimilation and its regulation in bacteria. *Annual review of microbiology* vol. 30 535–578 Preprint at <https://doi.org/10.1146/annurev.mi.30.100176.002535> (1976).
22. Bumann, D. & Schothorst, J. Intracellular *Salmonella* metabolism. *Cell Microbiol* **19**, e12766 (2017).
23. Choi, S. *et al.* The *Salmonella* virulence protein MgtC promotes phosphate uptake inside macrophages. *Nature Communications* 2019 10:1 **10**, 1–14 (2019).
24. Yang, W. *et al.* Phosphate (Pi) Transporter PIT1 Induces Pi Starvation in *Salmonella*-Containing Vacuole in HeLa Cells. *Int J Mol Sci* **24**, 17216 (2023).
25. Röder, J., Felgner, P. & Hensel, M. Single-cell analyses reveal phosphate availability as critical factor for nutrition of *Salmonella enterica* within mammalian host cells. *Cell Microbiol* **23**, e13374 (2021).
26. MacLean, M. J., Ness, L. S., Ferguson, G. P. & Booth, I. R. The role of glyoxalase I in the detoxification of methylglyoxal and in the activation of the KefB K⁺ efflux system in *Escherichia coli*. *Mol Microbiol* **27**, 563–571 (1998).
27. Ozyamak, E. *et al.* The critical role of S-lactoylglutathione formation during methylglyoxal detoxification in *Escherichia coli*. *Mol Microbiol* **78**, 1577–1590 (2010).

28. Clugston, S. L. *et al.* Isolation and sequencing of a gene coding for glyoxalase I activity from *Salmonella typhimurium* and comparison with other glyoxalase I sequences. *Gene* **186**, 103–111 (1997).
29. Hopper, D. J. & Cooper, R. A. The purification and properties of *Escherichia coli* methylglyoxal synthase. *Biochemical Journal* **128**, 321–329 (1972).
30. Subedi, K. P., Kim, I., Kim, J., Min, B. & Park, C. Role of GldA in dihydroxyacetone and methylglyoxal metabolism of *Escherichia coli* K12. *FEMS Microbiol Lett* **279**, 180–187 (2008).
31. Lee, C., Kim, I. & Park, C. Glyoxal detoxification in *Escherichia coli* K-12 by NADPH dependent aldo-keto reductases. *Journal of Microbiology* **51**, 527–530 (2013).
32. Kant, S., Liu, L. & Vazquez-Torres, A. The methylglyoxal pathway is a sink for glutathione in *Salmonella* experiencing oxidative stress. *PLoS Pathog* **19**, e1011441 (2023).
33. Barthel, M. *et al.* Pretreatment of mice with streptomycin provides a *Salmonella enterica* serovar Typhimurium colitis model that allows analysis of both pathogen and host. *Infect Immun* **71**, 2839–2858 (2003).
34. Diard, M. M. *et al.* Stabilization of cooperative virulence by the expression of an avirulent phenotype. *Nature* **494**, 353–356 (2013).
35. De Paepe, M. *et al.* Trade-Off between Bile Resistance and Nutritional Competence Drives *Escherichia coli* Diversification in the Mouse Gut. *PLoS Genet* **7**, e1002107- (2011).
36. Sousa, A. *et al.* Recurrent Reverse Evolution Maintains Polymorphism after Strong Bottlenecks in Commensal Gut Bacteria. *Mol Biol Evol* **34**, 2879–2892 (2017).
37. Cherrak, Y. *et al.* Non-canonical start codons confer context-dependent advantages in carbohydrate utilization for commensal *E. coli* in the murine gut. *Nat Microbiol* (2024) doi:10.1038/s41564-024-01775-x.
38. Garima, B., Mostafa, G., T., S. K., E., G. J. & M., T. S. Genetic engineering of *Salmonella* spp. for novel vaccine strategies and therapeutics. *EcoSal Plus* **0**, eesp-0004-2023 (2024).
39. Takaya, A., Yamamoto, T. & Tokoyoda, K. Humoral Immunity vs. *Salmonella*. *Front Immunol* **10**, (2020).
40. Baliban, S. M., Lu, Y.-J. & Malley, R. Overview of the Nontyphoidal and Paratyphoidal *Salmonella* Vaccine Pipeline: Current Status and Future Prospects. *Clinical Infectious Diseases* **71**, S151–S154 (2020).
41. Sears, K. T., Galen, J. E. & Tennant, S. M. Advances in the development of *Salmonella*-based vaccine strategies for protection against Salmonellosis in humans. *J Appl Microbiol* **131**, 2640–2658 (2021).

42. Diard, M. *et al.* A rationally designed oral vaccine induces immunoglobulin A in the murine gut that directs the evolution of attenuated *Salmonella* variants. *Nat Microbiol* **6**, 830–841 (2021).
43. Lentsch, V. *et al.* “EvoVax” – A rationally designed inactivated *Salmonella* Typhimurium vaccine induces strong and long-lasting immune responses in pigs. *Vaccine* **41**, 5545–5552 (2023).
44. Barroso-Batista, J. *et al.* The First Steps of Adaptation of *Escherichia coli* to the Gut Are Dominated by Soft Sweeps. *PLoS Genet* **10**, e1004182 (2014).
45. Eberl, C. *et al.* *E. coli* enhance colonization resistance against *Salmonella* Typhimurium by competing for galactitol, a context-dependent limiting carbon source. *Cell Host Microbe* **29**, 1680-1692.e7 (2021).
46. Oliveira, R. A. *et al.* *Klebsiella michiganensis* transmission enhances resistance to Enterobacteriaceae gut invasion by nutrition competition. *Nat Microbiol* **5**, 630–641 (2020).
47. Prax, N. *et al.* A diet-specific microbiota drives *Salmonella* Typhimurium to adapt its in vivo response to plant-derived substrates. *Anim Microbiome* **3**, 24 (2021).
48. Qiwei, C. *et al.* *Salmonella* Typhimurium alters galactitol metabolism under ciprofloxacin treatment to balance resistance and virulence. *J Bacteriol* **206**, e00178-24 (2024).
49. Gül, E. *et al.* Differences in carbon metabolic capacity fuel co-existence and plasmid transfer between *Salmonella* strains in the mouse gut. *Cell Host Microbe* **31**, 1140-1153.e3 (2023).
50. Cooper, V. S. & Lenski, R. E. The population genetics of ecological specialization in evolving *Escherichia coli* populations. *Nature* **407**, 736–739 (2000).
51. MacLean, R. C., Bell, G. & Rainey, P. B. The evolution of a pleiotropic fitness tradeoff in *Pseudomonas fluorescens*. *Proceedings of the National Academy of Sciences* **101**, 8072–8077 (2004).
52. Yang, Y. *et al.* Within-host evolution of a gut pathobiont facilitates liver translocation. *Nature* **607**, 563–570 (2022).
53. Rita, F., G., W. K., W., H. D. & Sophie, H. Identification of *Salmonella* Pathogenicity Island-2 Type III Secretion System Effectors Involved in Intramacrophage Replication of *S. enterica* Serovar Typhimurium: Implications for Rational Vaccine Design. *mBio* **4**, 10.1128/mbio.00065-13 (2013).
54. Saliba, A.-E. *et al.* Single-cell RNA-seq ties macrophage polarization to growth rate of intracellular *Salmonella*. *Nat Microbiol* **2**, 16206 (2016).

55. Fields, P. I., Swanson, R. V, Haidaris, C. G. & Heffron, F. Mutants of Salmonella typhimurium that cannot survive within the macrophage are avirulent. *Proceedings of the National Academy of Sciences* **83**, 5189–5193 (1986).
56. Muller, A. J. *et al.* Salmonella gut invasion involves TTSS-2-dependent epithelial traversal, basolateral exit, and uptake by epithelium-sampling lamina propria phagocytes. *Cell Host Microbe* **11**, 19–32 (2012).
57. Fattinger, S. A. *et al.* Gasdermin D is the only Gasdermin that provides protection against acute Salmonella gut infection in mice. *Proceedings of the National Academy of Sciences* **120**, e2315503120 (2023).
58. Zhang, H. *et al.* YaeB, Expressed in Response to the Acidic pH in Macrophages, Promotes Intracellular Replication and Virulence of Salmonella Typhimurium. *International Journal of Molecular Sciences* vol. 20 Preprint at <https://doi.org/10.3390/ijms20184339> (2019).
59. Konings, W. N. & Rosenberg, H. Phosphate transport in membrane vesicles from Escherichia coli. *Biochim Biophys Acta* **508**, 370–378 (1978).
60. Rosenberg, H., Gerdes, R. G. & Chegwidan, K. Two systems for the uptake of phosphate in Escherichia coli. *J Bacteriol* **131**, 505–511 (1977).
61. van Veen, H. W., Abee, T., Kortstee, G. J. J., Konings, W. N. & Zehnder, A. J. B. Translocation of metal phosphate via the phosphate inorganic transport system of Escherichia coli. *Biochemistry* **33**, 1766–1770 (1994).
62. Sweet, G. *et al.* Glycerol facilitator of Escherichia coli: cloning of glpF and identification of the glpF product. *J Bacteriol* **172**, 424 LP – 430 (1990).
63. Sweet, G. *et al.* Glycerol facilitator of Escherichia coli: cloning of glpF and identification of the glpF product. *J Bacteriol* **172**, 424 LP – 430 (1990).
64. Thauer, R. K., Jungermann, K. & Decker, K. Energy conservation in chemotrophic anaerobic bacteria. *Bacteriol Rev* **41**, 100–180 (1977).
65. Wang, B., Grant, R. A. & Laub, M. T. ppGpp Coordinates Nucleotide and Amino-Acid Synthesis in E. coli During Starvation. *Mol Cell* **80**, 29-42.e10 (2020).
66. Cummings, L. A., Wilkerson, W. D., Bergsbaken, T. & Cookson, B. T. In vivo, fliC expression by Salmonella enterica serovar Typhimurium is heterogeneous, regulated by ClpX, and anatomically restricted. *Mol Microbiol* **61**, 795–809 (2006).
67. Parys, K. *et al.* Signatures of antagonistic pleiotropy in a bacterial flagellin epitope. *Cell Host Microbe* **29**, 620-634.e9 (2021).

68. Hauser, E., Junker, E., Helmuth, R. & Malorny, B. Different mutations in the *oafA* gene lead to loss of O5-antigen expression in *Salmonella enterica* serovar Typhimurium. *J Appl Microbiol* **110**, 248–253 (2011).
69. Slauch, J. M., Lee, A. A., Mahan, M. J. & Mekalanos, J. J. Molecular characterization of the *oafA* locus responsible for acetylation of *Salmonella typhimurium* O-antigen: *oafA* is a member of a family of integral membrane trans-acylases. *J Bacteriol* **178**, 5904–5909 (1996).
70. Gül, E. *et al.* *Salmonella* T3SS-2 virulence enhances gut-luminal colonization by enabling chemotaxis-dependent exploitation of intestinal inflammation. *Cell Rep* **43**, 113925 (2024).